# SYMMETRIC BEHAVIOR REGULARIZED POLICY OPTIMIZATION

## ABSTRACT

Behavior Regularized Policy Optimization (BRPO) leverages asymmetric (divergence) regularization to mitigate the distribution shift in offline Reinforcement Learning. This paper is the first to study the open question of symmetric regularization. We show that symmetric regularization does not permit an analytic optimal policy $\pi^*$, posing a challenge to practical utility of symmetric BRPO. We approximate $\pi^*$ by the Taylor series of Pearson-Vajda $\chi^n$ divergences and show that an analytic policy expression exists only when the series is capped at $n = 5$. To compute the solution in a numerically stable manner, we propose to Taylor expand the conditional symmetry term of the symmetric divergence loss, leading to a novel algorithm: Symmetric $f$-Actor Critic (S$f$-AC). S$f$-AC achieves consistently strong results across various D4RL MuJoCo tasks. Additionally, S$f$-AC avoids per-environment failures observed in IQL, SQL, XQL and AWAC, opening up possibilities for more diverse and effective regularization choices for offline RL.

## 1 INTRODUCTION

Behavior regularized policy optimization (BRPO) is a simple, yet effective method that has attracted significant research interest for offline reinforcement learning (RL). By regularizing towards the behavior policy via a divergence penalty, BRPO effectively suppresses distributional shift incurred by out-of-distribution actions that may have spuriously high values. While many works have studied the sensitivity of BRPO to the behavior policy (Kostrikov et al., 2022; Ma et al., 2024), this paper investigates an orthogonal direction by focusing on the role played by distinct divergence regularizers.

Many BRPO algorithms share the following two steps:

$$\pi^* = \arg\max_{\pi} \ \mathbb{E}_{s \sim \mathcal{D}, a \sim \pi} \left[ Q(s, a) - \tau D_{\text{Reg}}(\pi(\cdot|s) \, \| \, \pi_{\mathcal{D}}(\cdot|s)) \right], \tag{1}$$

$$\pi_\theta = \arg\min_{\theta} \ \mathbb{E}_{s \sim \mathcal{D}} \left[ D_{\text{Opt}}(\pi^*(\cdot|s) \, \| \, \pi_\theta(\cdot|s)) \right]. \tag{2}$$

where $\mathcal{D}$ denotes the dataset, $\pi^*$ the optimal policy, $\pi_{\mathcal{D}}$ the behavior policy, and $\pi_\theta$ a parametrized policy that is being optimized. Here, Eq. (1) defines the theoretical maximizer of the regularized objective. However, this maximizer is not practical since computing $\pi^*$ is generally intractable. To deal with this issue, Eq. (2) defines an approximation that can be used in practice. $D_{\text{Reg}}$ and $D_{\text{Opt}}$ are most commonly defined using a KL divergence and their properties as such are well studied in the literature (Wu et al., 2020; Jaques et al., 2020; Chan et al., 2022). Some other well-studied candidates include the $\alpha$- and Tsallis divergences (Xu et al., 2023; Zhu et al., 2023). All of these divergences belong to the asymmetric class, e.g. $D_{\text{Reg}}(\pi\|\mu) \neq D_{\text{Reg}}(\mu\|\pi)$ for non-identical policies $\pi, \mu$.

There are two primary motivations for our work. First, using a symmetric divergence for $D_{\text{Reg}}$ has not been previously studied, nor is it known what is the corresponding symmetry-regularized optimal policy. Recent investigations have shown the benefits of defining $D_{\text{Opt}}$ using symmetric divergences: improved performance is observed in alignment of LLM than the standard KL (Wen et al., 2023; Go et al., 2023; Wang et al., 2024).

Second, we want to derive algorithms that have $D_{\text{Reg}} = D_{\text{Opt}}$, because we can get suboptimal solutions when there is an inconsistency between the two choices of regularizers. The inconsistency arises because the regularizer $D_{\text{Reg}}$ defines a unique regularized optimal policy. If the parametric policy $\pi_\theta$ is not optimized with the same divergence, it will be biased towards the optimum of

$D_{\text{Opt}}$ rather than $D_{\text{Reg}}$, incurring extra policy error and altering policy improvement dynamics, see Appendix B.3 for detailed discussion. Therefore, a key motivation of this paper lies in improving existing work (Go et al., 2023; Wang et al., 2024) that have KL as its $D_{\text{Reg}}$ but symmetric divergence as its optimization objective $D_{\text{Opt}}$. We achieve this by analyzing how symmetric divergences can be utilized in a principled manner in both $D_{\text{Reg}}$ and $D_{\text{Opt}}$, bringing maximal consistency.

Given these motivations, we aim to address the following key question in this paper:

*Does symmetric regularization permit an analytic optimal policy $\pi^*$?*

In other words, we want to know whether the policy $\pi^*$ can be expressed using elementary functions. It is possible that some $D_{\text{Reg}}$ induce optimal policies $\pi^*$ that are non-analytic in nature. However, it is vital that $\pi^*$ possess an analytic expression in order for the optimization Eq.(2) to be usable in practice. In this paper we are interested in the case where $D_{\text{Reg}}$ and $D_{\text{Opt}}$ use the same symmetric divergences for maximal consistency.

Unfortunately, the answer to this question is *no*. Although we can formulate symmetric regularization using the $f$-divergence framework, it does not permit an analytic solution in general. To address this challenge, we take advantage of the fact that the Taylor expansion of any $f$-divergence is an infinite series in the Pearson-Vajda $\chi^n$-divergences (Nielsen & Nock, 2013). We prove that analytic $\pi^*$ can be obtained when this series is truncated to finite. We call the proposed algorithm: Symmetric $f$-Actor-Critic (S$f$-AC). Table 1 lists all the symmetric divergences that are considered in this paper. To the best of our knowledge, our work is the first to study symmetric BRPO and to derive a corresponding practical algorithm.

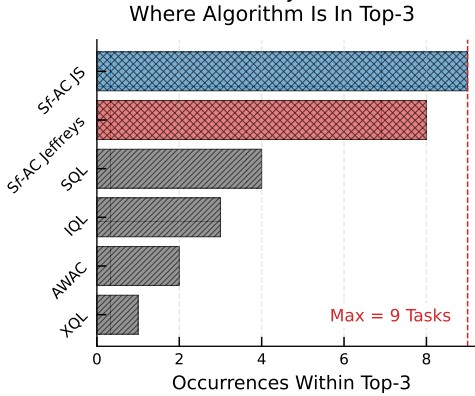

Figure 1: Number of times each algorithm is amongst the top-3 performers on 9 D4RL MuJoCo tasks. Our method (S$f$-AC) remains relatively more stable across tasks compared to the baselines.

We verify that S$f$-AC performs competitively on a distribution approximation task and the standard D4RL MuJoCo offline benchmark, opening up new research possibilities for BRPO. Figure 1 summarizes the benefit of using S$f$-AC on D4RL MuJoCo. The bars count the number of times an algorithm's performance is amongst the top-3 based on AUC on 9 D4RL MuJoCo tasks (3 environments with 3 difficulties). We observe that S$f$-AC Jensen-Shannon and Jeffreys are less prone to per-environment failures that are encountered by other algorithms. Additionally, **S$f$-AC Jensen-Shannon is the only algorithm which consistently ranked within top-3 across all the tested tasks.** See Figure 3 for in-depth results.

## 2 PRELIMINARY

### 2.1 OFFLINE REINFORCEMENT LEARNING

We focus on discounted Markov Decision Processes (MDPs) expressed by the tuple $(\mathcal{S}, \mathcal{A}, P, r, \gamma)$, where $\mathcal{S}$ and $\mathcal{A}$ denote state space and action space, respectively. $P$ denotes the transition probability, $r$ defines the reward function, and $\gamma \in (0, 1)$ is the discount factor. A policy $\pi$ is a mapping from the state space to distributions over actions. We define the action value and state value as $Q^\pi(s, a) = \mathbb{E}_\pi \left[ \sum_{t=0}^\infty \gamma^t r(s_t, a_t) | s_0 = s, a_0 = a \right]$, $V^\pi(s) = \mathbb{E}_\pi [Q^\pi(s, a)]$. In this paper we focus on the offline RL context where the agent learns from a static dataset $\mathcal{D} = \{s_i, a_i, r_i, s_i'\}$ storing transitions. We denote $\pi_\mathcal{D}$ by the behavior policy that generated the dataset $\mathcal{D}$.

Behavior regularized policy optimization (BRPO) is a simple yet effective method for offline RL. It solves the following objective:

$$\max_\pi \; \mathbb{E}_{\substack{s \sim \mathcal{D} \\ a \sim \pi}} \left[ Q(s, a) - \tau D_{\text{Reg}}(\pi(\cdot|s) || \pi_\mathcal{D}(\cdot|s)) \right], \tag{3}$$

Brute-force solving Eq. (3) is intractable for high-dimensional continuous action spaces as it entails solving the integral over actions. Therefore, Eq.(3) is translated into Eqs.(1), (2) in the introduction.

The functional form of $\pi^*$ can be determined by the convexity of $D_{\text{Reg}}$ (Hiriart-Urruty & Lemaréchal, 2004), but it is generally intractable to sample from it due to the normalization constant. Instead, an easy-to-sample surrogate policy $\pi_\theta$ is computed from $\pi^*$ by minimizing $D_{\text{Reg}}$. The surrogate can be considered as an approximate maximizer to the original objective.

## 2.2 $f$-DIVERGENCES AND ASYMMETRIC BRPO

A $f$-divergence that measures the difference between two continuous policies $\pi, \mu$ is defined as follows (Ali & Silvey, 1966; Sason & Verdú, 2016).

**Definition 1.** The $f$-divergence from policy $\mu$ to $\pi$ is defined by:

$$D_f(\pi(\cdot|s)\,||\,\mu(\cdot|s)) := \int_{\mathcal{A}} \mu(a|s) f\left(\frac{\pi(a|s)}{\mu(a|s)}\right) \mathrm{d}a = \mathbb{E}_\mu\left[f\left(\frac{\pi(a|s)}{\mu(a|s)}\right)\right].$$

where $f$ is a convex, lower semi-continuous function satisfying $f(1) = 0$ and we have assumed that $\pi$ is absolutely continuous with respect to $\mu$.

Framing BRPO using $f$-divergence helps us choose $D_{\text{Reg}}$ by verifying if its $f$ satisfies the conditions. For instance, KL BRPO corresponds to $f(t) = t \ln t$ and it leads to $\pi^*(a|s) \propto \pi_{\mathcal{D}}(a|s) \exp\left(\tau^{-1}\left(Q(s,a) - V(s)\right)\right)$. Though this policy is intractable to directly compute due to its normalization constant, it is known that by opting for the same KL as $D_{\text{Opt}}$ in Eq. (2) yields

$$\pi_\theta = \arg\min_\theta \; \mathbb{E}_{(s,a)\sim\mathcal{D}}\left[-\exp\left(\frac{Q(s,a) - V(s)}{\tau}\right) \ln \pi_\theta(a|s)\right]. \tag{4}$$

This method called advantage regression has been extensively applied in the offline context (Peng et al., 2020; Jaques et al., 2020; Garg et al., 2023). Similarly, we can validate symmetric BRPO provided that their $f$ functions satisfy the conditions and have an analytic solution $\pi^*$. In next section we define symmetric divergences.

## 2.3 ISSUE WITH ASYMMETRIC $D_{\text{REG}}$, SYMMETRIC $D_{\text{OPT}}$

Existing methods (Go et al., 2023; Wang et al., 2024) have opted for symmetric divergences as $D_{\text{Opt}}$. We argue this is suboptimal by revisiting the following analysis. First, the policy improvement step is performed:

$$\pi^* = \arg\max_\pi \; \mathbb{E}_{a\sim\pi}[Q(s,a)] - \tau D_{\text{Reg}}(\pi(\cdot|s)||\pi_{\mathcal{D}}(\cdot|s))$$

Here, $D_{\text{Reg}}$ directly shapes the optimality condition and the shape of the target distribution itself. Therefore, introducing symmetry into $D_{\text{Reg}}$ changes how the policy improves, changing the weightings of different actions within the support of the behavior dataset. However, existing methods still adopt asymmetric $D_{\text{Reg}}$ due to the intractability. Rather, they have opted for a symmetric loss $D_{\text{Opt}}$, which corresponds to the projection step: $\pi_\theta = \arg\min_\theta D_{\text{Opt}}(\pi(\cdot|s)||\pi_{\mathcal{D}}(\cdot|s))$.

If $D_{\text{Opt}} \neq D_{\text{Reg}}$, the resulting policy $\pi_\theta$ is the optimal fit for the target $\pi^*$ according to the metric $D_{\text{Opt}}$, but not $D_{\text{Reg}}$ that defined the original regularization problem. **This means $\pi_\theta$ and the $Q$-function $Q(s,a)$ will be inconsistent with the true regularization objective,** and the fixed point achieved by the mismatched iteration will be biased away from the true solution $\pi^*$, degrading performance and theoretical guarantees, see Appendix B.3 for detail.

# 3 ISSUES WITH SYMMETRIC BRPO

This section investigates the issues with symmetric BRPO. We formally define symmetric divergences of interest in this paper (Definition 2, 3). We then show that BRPO with the given symmetric divergences does not permit any analytic policy solution (Theorem 1). Finally, symmetric divergences as optimization objective can incur numerical issues (Theorem 2).

**Definition 2** (Pardo (2006))**.** Assume that the $f$ function satisfies the conditions in Definition 1, then $f(t) + tf\left(\frac{1}{t}\right)$ defines a symmetric divergence.

| | Divergence | $D_f(\pi\|\mu)$ | $f(t)$ |
|---|---|---|---|
| Asymmetric | Forward KL | $\int \pi(x) \ln \frac{\pi(x)}{\mu(x)}\,\mathrm{d}x$ | $t \ln t$ |
| | Reverse KL | $\int \mu(x) \ln \frac{\mu(x)}{\pi(x)}\,\mathrm{d}x$ | $-\ln t$ |
| Symmetric | Jeffrey | $\int (\pi(x) - \mu(x)) \ln \left(\frac{\pi(x)}{\mu(x)}\right)\,\mathrm{d}x$ | $(t-1)\ln t$ |
| | Jensen-Shannon | $\frac{1}{2}\int \pi(x) \ln \frac{2\pi(x)}{\pi(x)+\mu(x)} + \mu(x) \ln \frac{2\mu(x)}{\pi(x)+\mu(x)}\,\mathrm{d}x$ | $t \ln t - (1+t)\ln\frac{1+t}{2}$ |
| | GAN divergence | $\int \pi(x) \ln \frac{2\pi(x)}{\pi(x)+\mu(x)} + \mu(x) \ln \frac{2\mu(x)}{\pi(x)+\mu(x)}\,\mathrm{d}x - \ln(4)$ | $t \ln t - (1+t)\ln(1+t)$ |

Table 1: Asymmetric and symmetric $f$-divergences, their definitions and the generator functions. Jeffrey's divergence equals forward KL adds reverse KL. GAN divergence refers to the modified Jensen-Shannon divergence used by Generative Adversarial Nets (Goodfellow et al., 2014).

Pardo's definition identifies exactly one symmetric divergence given $f$, i.e. once $f$ is specified the symmetric divergence is also determined. However, in the machine learning context, researchers are more interested in the following more flexible definition (Nowozin et al., 2016).

**Definition 3** (Sason & Verdú (2016)). We consider symmetric divergences defined by $f(t) = t \ln t + g(t)$ where $g$ is an arbitrary convex function called *conditional symmetry function*.

Sason & Verdú's definition is broad enough to cover all the symmetric divergences of interest in the machine learning community (Sason & Verdú, 2016; Nowozin et al., 2016). These divergences are listed in Table 1. In this paper, we follow Sason & Verdú's definition and answer the key question posted in Introduction: symmetric regularization does not permit an analytic optimal solution $\pi^*$, unlike the popular asymmetric counterpart.

**Theorem 1.** *Let the symmetric $f$-divergence be defined by Definition 3. Then if $g'(t)$ does not make $f'(t)$ an affine function in $\ln t$, i.e. $f'(t) \neq a \ln t + b$, the regularized optimal policy $\pi^*$ does not have an analytic expression.*

*Proof.* See Appendix A.1 for a proof. □

It is worth noting that this affine function should be strictly in $\ln t$, excluding terms like $\ln(t+1)$. As an immediate consequence, the symmetric divergences in Table 1 do not have an analytic $\pi^*$.

**Corollary 1.** *The Jeffrey's, Jensen-Shannon and GAN divergence do not permit an analytic $\pi^*$.*

Corollary 1 can be verified by checking that the Jeffrey's divergence has $g'(t) = -\frac{1}{t}$, the Jensen-Shannon $g'(t) = -\ln\frac{t+1}{2} - 1$, and the GAN $g'(t) = -\ln(t+1) - 1$.

Characterizing the solution of $f$-divergence regularization has been discussed extensively in both BRPO and the distribution correction estimation (DICE) literature (Nachum et al., 2019; Nachum & Dai, 2020; Lee et al., 2021). In Appendix A.4 we show their characterizations lead to the same conclusion. These methods focused exclusively on asymmetric candidates such as the $\chi^2$ or KL divergence. We are not aware of any published result that discussed symmetric regularization.

We next examine symmetric divergences as optimization objective Eq.(2). It turns out they can also be numerically problematic, especially for finite support policies (i.e. $\exists a, \pi(a|s) = 0$) that can be induced by popular asymmetric divergences like the Tsallis or $\alpha$-divergences (Li et al., 2023; Xu et al., 2023; Zhu et al., 2025a) that attract increasing interest in recent offline RL studies.

**Theorem 2.** *Minimizing symmetric divergence losses in Eq. (2) can incur numerical issues when either $\pi^*$ or $\pi_\theta$ has finite support.*

*Proof.* This result is a natural consequence of symmetry. Consider the Jeffrey's divergence (Table 1) for instance, a finite support policy must incur division by zero. □

It is worth noting that even for full-support policies, the issue can persist when $\pi^* \gg \pi_\theta$ and vice versa. While some candidates like the Jensen-Shannon may not necessarily incur the same issue, existing research has reported instability (Go et al., 2023). It is therefore important to find new tools to circumvent vanilla symmetric divergences.

## 4 SYMMETRIC BRPO VIA TAYLOR EXPANSION

Our proposed method centers on the Taylor's series so that it permits both an analytic policy and stable optimization.

### 4.1 SYMMETRIC REGULARIZATION VIA THE $\chi^n$ SERIES

To address the issues of symmetric divergences, we recall the following classic result establishing equivalence between a $f$-divergence and its Taylor series.

**Lemma 3** ((Nielsen & Nock, 2013)). *On top of Definition 1, further let $f$ be $n$-times differentiable. Then a valid $f$-divergence permits the following Taylor expansion*

$$D_f\left(\pi(\cdot|s)||\pi_{\mathcal{D}}(\cdot|s)\right) = \int \pi_{\mathcal{D}}(a|s) \sum_{n=0}^{\infty} \frac{f^{(n)}(1)}{n!} \left(\frac{\pi(a|s)}{\pi_{\mathcal{D}}(a|s)} - 1\right)^n \mathrm{d}a,$$

*where $f^{(n)}$ denotes the $n$-th order derivative of $f$. Recognize that $\int \pi_{\mathcal{D}}(a|s)(\frac{\pi(a|s)}{\pi_{\mathcal{D}}(a|s)} - 1)^n \mathrm{d}a = \chi^n\left(\pi(\cdot|s)||\pi_{\mathcal{D}}(\cdot|s)\right)$ is the Pearson-Vajda $\chi^n$ divergence.*

It is perhaps surprising that though $\chi^n$ divergence is asymmetric, Lemma 3 establishes an equivalence between any symmetric $f$-divergence and the infinite series in $\chi^n$. The equivalence allows us to formulate symmetric BRPO as:

$$\max_{\pi} \mathbb{E}_{\substack{s\sim\mathcal{D} \\ a\sim\pi}} \left[Q(s,a) - \tau \sum_{n=0}^{N} \frac{f^{(n)}(1)}{n!} \chi^n\left(\pi(\cdot|s)||\pi_{\mathcal{D}}(\cdot|s)\right)\right], \tag{5}$$

When $N = \infty$, we recover Eq. (1) with arbitrary valid $f$-divergence. However, we again face the same issue of no analytic solution under symmetry. To this end, we truncate the series to $N < \infty$ to retreat from exact symmetry. The following result, building on the famous Abel-Ruffini theorem (Ramond, 2022) shows that an analytic solution is available only when $N < 5$.

**Theorem 3.** *Let the series in Eq.(5) be truncated to $N$, i.e. $n = 0, 1, \ldots, N$ with $2 \leq N < 5$. Then the regularized optimal policy $\pi^*$ can be expressed analytically as*

$$\pi^*(a|s) \propto \pi_{\mathcal{D}}(a|s) \left[1 + Z_N(s,a)\right]_+,$$

*where $Z_N(s,a)$ contains $N$-radicals of $Q(s,a)$ and $[\cdot]_+ := \max\{\cdot, 0\}$. Moreover, when $N = 2$:*

$$\pi^*(a|s) \propto \pi_{\mathcal{D}}(a|s) \left[1 + \frac{Q(s,a)}{\tau}\right]_+, \tag{6}$$

*where we absorbed additional constants to $\tau$. $N \geq 2$ because $f^{(0)}(1) = 0$ and $\chi^1 = 0$.*

*Proof.* See Appendix A.2 for a proof. □

Notice that $N = 2$ gives $D_f(\pi||\mu) \approx \frac{f''(1)}{2}\chi^2(\pi||\mu)$, with $f''(1) > 0$ by the strong convexity of $f$. When $f(t) = t \ln t$, it recovers the classic result that $\chi^2(\pi||\mu) \approx 2D_{\mathrm{KL}}(\pi || \mu)$. Generally, the series Eq.(8) allows one to balance between symmetric regularization and the truncation threshold $[Z_N]_+$. But as shown by Zhu et al. (2023), the threshold can also be fully controlled by $\tau$. Therefore, we can safely choose $N = 2$ which induces Eq.(6).

### 4.2 SYMMETRIC OPTIMIZATION BY EXPANDING CONDITIONAL SYMMETRY

The numerical instability of vanilla symmetric divergence could again be addressed by the Taylor expansion. However, a full Taylor expansion involves purely powers of policy ratio which can be numerically unstable for minimization. Instead, we draw a key observation from Definition 3 that the symmetric divergences can be decoupled into two interdependent terms $t \ln t$ and $g(t)$. Therefore, we can decompose the optimization step as the following:

$$\mathcal{L}(\theta) := \mathbb{E}_{s\sim\mathcal{D}} \left[D_{\mathrm{Opt}}\left(\pi^*(\cdot|s)||\pi_\theta(\cdot|s)\right)\right]$$

$$= \underbrace{\mathbb{E}_{s\sim\mathcal{D}} \left[D_{\mathrm{KL}}(\pi^*(\cdot|s) || \pi_\theta(\cdot|s))\right]}_{t \ln t} + \underbrace{\mathbb{E}_{s\sim\mathcal{D}} \left[\int \pi_\theta(a|s) \, g\left(\frac{\pi^*(a|s)}{\pi_\theta(a|s)}\right) \mathrm{d}a\right]}_{\text{conditional symmetry}}. \tag{7}$$

| $D_{\text{Opt}}$ | $f(t)$ | $g(t)$ | $g^{(n)}(1), (n \geq 2)$ | Series coefficient |
|---|---|---|---|---|
| Jeffrey | $(t-1)\ln t$ | $-\ln t$ | $(-1)^n (n-1)!$ | $\sum_{n=2}^{\infty} (-1)^n \frac{1}{n}$ |
| Jensen-Shannon | $t\ln t - (1+t)\ln\frac{1+t}{2}$ | $-(1+t)\ln\frac{1+t}{2}$ | $(-1)^n (n-2)! \frac{1}{2^{n-1}}$ | $\sum_{n=2}^{\infty} \frac{(-1)^n}{n(n-1)2^{n-1}}$ |
| GAN Divergence | $t\ln t - (1+t)\ln(1+t)$ | $-(1+t)\ln(1+t)$ | $(-1)^n (n-2)! \frac{1}{2^{n-1}}$ | $\sum_{n=2}^{\infty} \frac{(-1)^n}{n(n-1)2^{n-1}}$ |

Table 2: Symmetric divergences, their $f$ generators, conditional symmetry $g$, derivatives and Taylor expansion series coefficients. JS and GAN share the same derivatives and series coefficients.

The first term $t\ln t$ corresponds to the advantage regression in Eq.(4). The second term that may vary case by case is called the conditional symmetry term. Note that $\pi^*$ can be zero for some actions. To ensure the second term is valid, it is required that for actions sampled from $\pi_\theta$ the function $g$ cannot involve terms that flip the ratio e.g. $-\ln t$ that destroys the validity.

To this end, we Taylor-expand the conditional symmetry term $g(t)$ to obtain our final loss objective:

$$\mathcal{L}(\theta) = \mathbb{E}_{(s,a)\sim\mathcal{D}} \left[ -\left[1 + \frac{Q(s,a)-V(s)}{\tau}\right]_+ \ln\pi_\theta(a|s)\right] + \mathbb{E}_{\substack{s\sim\mathcal{D}\\a\sim\pi_\theta}}\left[\sum_{n=2}^{N_{\text{loss}}} \frac{f^{(n)}(1)}{n!}\left(\frac{\pi^*(a|s)}{\pi_\theta(a|s)} - 1\right)^n\right]. \tag{8}$$

Table 2 summarizes the series coefficients. Compared to full expansion, expanding only the conditional symmetry part has an additional benefit: for large $n$ with high order policy ratio, its coefficient decays quickly towards zero and lowers the importance of higher order terms.

Our method has an interesting connection to a very recent method (Huang et al., 2025) that proposed the KL + $\chi^2$ regularization to improve RLHF alignment. Despite different settings, Eq.(8) can be seen as a generalization of their method since we recover their method by truncating the series at $N_{\text{loss}} = 2$, with the coefficient $\frac{f^n(1)}{2}$ playing the role of tuning relative importance.

## 5 SYMMETRIC $f$-ACTOR-CRITIC

The Taylor series was expanded at around $t = 1$, suggesting that the policy ratio should stay in the neighborhood of 1 for the series to converge. To this end, we clip the ratio to the interval $[1-\epsilon, 1+\epsilon]$, $\epsilon > 0$. As such, Taylor expansion provides an interesting interpretation to the proximal policy optimization (PPO) style clipping (Schulman et al., 2017). In fact, we can connect the convergence requirements to empirical clipping which plays a key role in many PPO-type methods (Vaswani et al., 2022; Zhuang et al., 2023), see the related work section for detail. Moreover, we can show that the distance from the clipped truncated series to its exact counterpart is upper-bounded:

**Theorem 4.** *Assume the ratio $\pi^*/\pi_\theta$ is clipped to the interval $[1-\epsilon, 1+\epsilon]$, on which $g^{(n)}$ is absolutely continuous. Assume further the states are randomly sampled from the dataset. Let $\mathcal{L}^\infty$ denote the infinite series of $g(t)$ and $\widehat{\mathcal{L}}_\epsilon^N(\theta)$ the $N$-term truncated series with clipping, then*

$$\left|\mathcal{L}^\infty(\theta) - \widehat{\mathcal{L}}_\epsilon^N(\theta)\right| \leq \frac{2\epsilon^{N+1}}{(N+1)!}\left\|g^{(N+1)}\right\|_\infty,$$

*where $\left\|g^{(N+1)}\right\|_\infty := \sup_{t\in[1-\epsilon,1+\epsilon]}\left|g^{(N+1)}(t)\right|$.*

*Proof.* See Appendix A.3 for a proof. □

Theorem 4 shows that the actual loss objective is not far from the exact infinite Taylor series, and hence $D_{\text{Opt}}$ is not far from a symmetric divergence. Consider the Jeffrey's divergence where $g(t) = -\ln t$ and $\epsilon = 0.2, N = 5$, then the upper bound yields $8.13 \times 10^{-5}$.

To obtain a practical implementation, for simplicity we assume at every policy update the action value $Q_\psi$ parametrized by $\psi$ and state value $V_\phi$ parametrized by $\phi$ are available. They are trained by the standard critic learning procedures which will be detailed in the appendix.

We also need to be able to evaluate $\pi^*(a|s)$ for actions sampled from $\pi_\theta$. To do this, we can either approximately evaluate $\pi_\mathcal{D}(a|s)\left[1 + \frac{Q_\psi(s,a)-V_\phi(s)}{\tau}\right]_+$ without the normalization constant but require estimating $\pi_\mathcal{D}$, or we parametrize $\pi^*$ by another network $\zeta$ that is trained by advantage regression. We find the latter approach more performing and stable in general. Alg. 1 lists our algorithm Symmetric $f$-divergence Actor-Critic (S$f$-AC), where $[\cdot]_\epsilon := \texttt{clip}(\cdot, 1-\epsilon, 1+\epsilon)$.

## 6 EXPERIMENTS

In the experiments we aim to show that (i) the conditional symmetry expansion is a valid loss function; (ii) S$f$-AC can perform well on the standard offline benchmark. Section 6.1 verifies (i) and section 6.2 and 6.3 for (ii).

---

**Algorithm 1:** Symmetric $f$-Actor-Critic

**Input:** $\mathcal{D}, \tau > 0, N_\text{loss} \geq 2, \epsilon > 0$
**while** learning **do**
 sample $(s, a)$ from dataset $\mathcal{D}$ ;
 compute $Q_\psi(s,a)$ and $V_\phi(s)$;
 compute advantage regression $\mathcal{L}_{t\ln t} :=$
$$-\widehat{\mathbb{E}}_{s,a}\left[\left[1 + \frac{Q_\psi(s,a)-V_\phi(s)}{\tau}\right]_+ \ln\pi_\theta(a|s)\right];$$
 sample $b$ from $\pi_\theta$ ;
 compute truncated series $\mathcal{L}_g :=$
$$\widehat{\mathbb{E}}_{s,b}\left[\sum_{n=2}^{N_\text{loss}} \frac{f^{(n)}(1)}{n!}\left(\left[\frac{\pi_\zeta(b|s)}{\pi_\theta(b|s)}\right]_\epsilon - 1\right)^n\right];$$
 update $\theta$ by minimizing $\mathcal{L}_{t\ln t} + \mathcal{L}_g$;
 update $\zeta$ by minimizing $\mathcal{L}_{t\ln t}$;
**end**

---

### 6.1 DISTRIBUTION MATCHING

Every loss divergence in the ideal case should lead to the same optimal solution. As a sanity check, we first verify that our Taylor expansion loss Eq.(8) is valid, in that minimizing it produces similar results to the existing divergence losses. We follow the setting in (Nowozin et al., 2016) to learn a Gaussian $\pi_\theta$ with parameters $\theta = (\mu, \sigma)$ to approximate a univariate mixture of Gaussians. The mixture Gaussian is shown in Figure 2.

**Setup.** Learning is performed by minimizing the expanded objective Eq.(8) with $N_\text{loss} = 5$. The target policy is the mixture and $\theta$ a two-layer neural network of 64 hidden dimensions. We minimize the objective by SGD with learning rate 0.001, batch size 128 for 1000 update steps. The optimal Gaussian parameters $\mu, \sigma$ are obtained by numerical integration. We compare them to the parameters obtained by minimizing our objectives and the vanilla divergences.

|  | Method | Jeffrey (values) | Jeffery (abs. error) ↓ | JS (values) | JS (abs. error) ↓ |
|---|---|---|---|---|---|
| | Optimal | 0.0159 | – | 0.0368 | – |
| $D_f$ | Vanilla | 0.0157 | 0.0002 | 0.0682 | 0.0314 |
| | Ours | 0.0161 | 0.0002 | 0.0387 | 0.0019 |
| | Optimal | 0 | – | 0 | – |
| $\mu$ | Vanilla | 0.0067 | 0.0067 | −0.0778 | 0.0778 |
| | Ours | −0.0166 | 0.0166 | −0.0475 | 0.0475 |
| | Optimal | 2.2218 | – | 2.2868 | – |
| $\sigma$ | Vanilla | 2.2396 | 0.018 | 4.5559 | 2.2691 |
| | Ours | 2.4926 | 0.2707 | 2.5438 | 0.2570 |

Table 3: Loss objectives $D_f$ and corresponding fit parameters $\mu, \sigma$. Optimal: values given by numerical integration. Vanilla: minimized the vanilla $f$-divergence per definition. Ours: minimized our Taylor expansion loss for $N_\text{loss} = 5$. Our JS loss brings a better fit than minimizing the vanilla $f$-divergence.

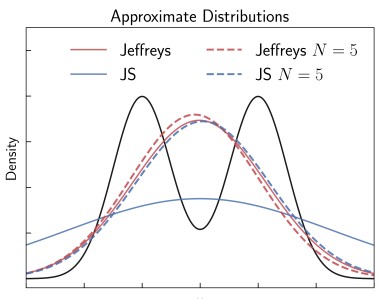

Figure 2: Approximating a mixture of Gaussians (black) by minimizing vanilla divergence (solid) and S$f$-AC loss for $N_\text{loss} = 5$ (dashed). Vanilla JS loss causes the Gaussian to lose track of optimal $\sigma^*$ given by numerical integration.

**Results.** Table 3 shows the divergence values and fit Gaussian parameters obtained by numerical integration (Optimal), Vanilla (vanilla $f$-divergence loss per definition) and our method (ours). It is visible that the S$f$-AC loss is a reasonable objective and induces consistent learned distribution behaviors (dashed curves) with the numerical solution. Moreover, as can be seen in Figure 2, minimizing the vanilla Jensen-Shannon (JS) divergence loss induces a much wider distribution (solid

blue) with $\sigma = 4.556$. The poor approximation coincides with the observation that exact symmetric divergence losses can lead to unstable policy behaviors (Wang et al., 2024).

## 6.2 MuJoCo

The D4RL MuJoCo suite has been a standard benchmark for testing various offline RL algorithms. In this section we compare S$f$-AC Jensen-Shannon and Jeffreys against the baselines on 9 environment-difficulty combination. We include AWAC that corresponds to explicit KL regularization (Nair et al., 2021), XQL to implicit KL regularization (Garg et al., 2023), SQL (Xu et al., 2023) to sparse $\alpha$-divergence regularization and IQL (Kostrikov et al., 2022). For S$f$-AC, we use $N_{\text{loss}} = 3, \epsilon = 100$ and perform grid search over $\tau$ and learning rates. For the baselines, we use the published settings, see Appendix B.1 for detail. All algorithms are run for $10^6$ steps and averaged over 5 seeds.

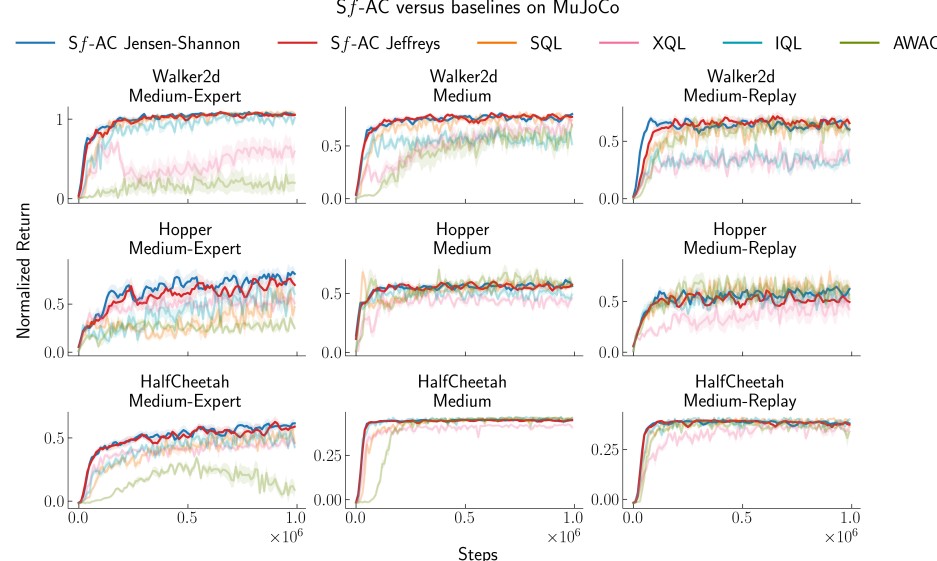

Figure 3: S$f$-AC Jensen-Shannon and Jeffreys with $N_{\text{loss}} = 3, \epsilon = 100$ versus baselines on the D4RL MuJoCo environments. Solid lines are mean and shaded regions the standard deviation, averaged over 5 seeds. Only S$f$-AC methods are shown with full opacity. Both Jensen-Shannon and Jeffrey's divergences performed favorably compared to the baselines.

Figure 3 shows the full result. Only S$f$-AC methods are shown with full opacity. Others are shown with transparency for uncluttered visualization. Except on Hopper medium and Hopper Medium-Replay, both symmetric divergences are among the top players on the rest of environments. Recall that XQL, SQL, IQL are very competent on the MuJoCo tasks, yet it is visible that S$f$-AC performs favorably against these methods. Table 4 confirms this observation. Moreover, Figure 8 demonstrates S$f$-AC is insensitive to the number of terms used to compute the symmetry divergence. This is desirable as S$f$-AC avoids the burdensome environment-specific parameter tuning to save resources and gain interpretability.

Table 4: Summary of D4RL results. Jeffrey and Jensen-Shannon are with $N = 3$.

| Environment | Dataset | Jeffrey | Jensen Shannon | AWAC | IQL | SQL | XQL |
|---|---|---|---|---|---|---|---|
| HalfCheetah | medexp | $0.5791 \pm 0.0265$ | $0.6048 \pm 0.0269$ | $0.0939 \pm 0.0542$ | $0.4856 \pm 0.0317$ | $0.5085 \pm 0.0460$ | $0.4665 \pm 0.0249$ |
| | medium | $0.4453 \pm 0.0030$ | $0.4433 \pm 0.0040$ | $0.4545 \pm 0.0062$ | $0.4517 \pm 0.0086$ | $0.4560 \pm 0.0019$ | $0.4178 \pm 0.0044$ |
| | medrep | $0.3791 \pm 0.0094$ | $0.3814 \pm 0.0079$ | $0.3408 \pm 0.0179$ | $0.3862 \pm 0.0127$ | $0.3873 \pm 0.0098$ | $0.3560 \pm 0.0130$ |
| Hopper | medexp | $0.7358 \pm 0.0455$ | $0.7853 \pm 0.0780$ | $0.2816 \pm 0.0355$ | $0.5935 \pm 0.0744$ | $0.5018 \pm 0.0932$ | $0.5383 \pm 0.0510$ |
| | medium | $0.5590 \pm 0.0199$ | $0.5891 \pm 0.0284$ | $0.5821 \pm 0.0292$ | $0.4732 \pm 0.0304$ | $0.5758 \pm 0.0348$ | $0.4306 \pm 0.0290$ |
| | medrep | $0.5196 \pm 0.0695$ | $0.5847 \pm 0.0691$ | $0.6144 \pm 0.0642$ | $0.5700 \pm 0.0671$ | $0.6285 \pm 0.0597$ | $0.4815 \pm 0.1144$ |
| Walker2d | medexp | $1.0518 \pm 0.0239$ | $1.0660 \pm 0.0193$ | $0.1972 \pm 0.1129$ | $1.0008 \pm 0.0374$ | $1.0858 \pm 0.0066$ | $0.6063 \pm 0.1257$ |
| | medium | $0.7741 \pm 0.0193$ | $0.7832 \pm 0.0156$ | $0.6054 \pm 0.0780$ | $0.5702 \pm 0.0337$ | $0.7601 \pm 0.0284$ | $0.6749 \pm 0.0240$ |
| | medrep | $0.6588 \pm 0.0422$ | $0.6394 \pm 0.0499$ | $0.5980 \pm 0.0559$ | $0.3498 \pm 0.0392$ | $0.6299 \pm 0.0438$ | $0.3700 \pm 0.0705$ |

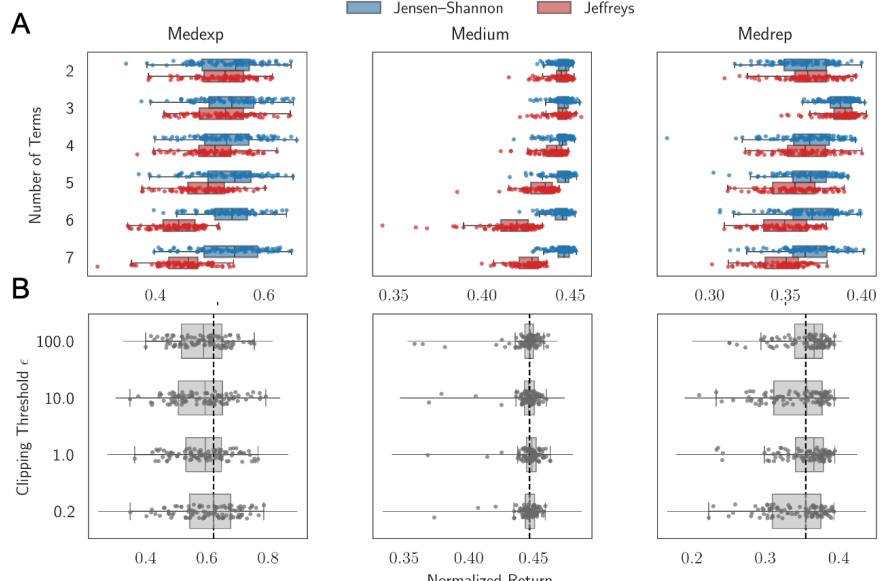

Figure 5: Ablation studies on HalfCheetah. (**A**) Scores of $Sf$-AC across $N_{\text{loss}}$ when $\epsilon = 100$. Scattered dots are evaluations from the last $20\%$ of learning. JS remains stable for large $N_{\text{loss}}$ as its series coefficients decay quickly to zero, see Table 2. By contrast, Jeffreys performance decreases as $N_{\text{loss}}$ increases. (**B**) Scores under various clipping thresholds $\epsilon$ of JS when $N_{\text{loss}} = 3$. Dashed vertical line shows the median performance of $\epsilon = 0.2$. Overall, $Sf$-AC is insensitive towards $\epsilon$.

Figure 4 compares policy evolution of $Sf$-AC Jensen-Shannon (JS) and AWAC. At step $0$ the policies are randomly initialized. Note that AWAC explicitly minimizes forward KL between $\pi^*$ and $\pi_\theta$. The allowed action range is $[-1, 1]$ but shown larger here for better visualization. It is visible that both JS and Jeffreys keep policies in the range, while the optimizing forward KL of AWAC increasingly prompts the policy beyond the allowed minimum action $-1$ (shaded area). Since actions outside the range are clipped (in this case $-1$), the policy will have an unintended shape and cause significant bias (Lee et al., 2025). Figure 9 in Appendix B.2 runs $Sf$-AC on the more challenging AntMaze, Franka Kitchen and Adroit Pen and confirms the same conclusion.

## 6.3 ABLATION STUDIES

From Alg. 1 it is visible that $Sf$-AC depends on two other hyperparameters: the number of terms for the conditional symmetry expansion $N_{\text{loss}}$ and the clipping threshold $\epsilon$. The former controls approximation to the symmetric divergence, and the latter controls the convergence of Taylor series. In this section we examine their effect in detail. Due to the page limit, we include additional results on generalized policy in Appendix B.2.

**Number of Terms $N_{\text{loss}}$.** We sweep $Sf$-AC Jensen-Shannon (JS) and Jeffreys over $N_{\text{loss}} = 2$ to $7$ on HalfCheetah. Figure 5 (A) shows the last $20\%$ of learning. Note that $N_{\text{loss}} = 2$ corresponds to $\chi^2$ divergence. It can be seen that the JS remains stable across $N_{\text{loss}}$, while the Jeffreys performance decreases along with the increase of $N_{\text{loss}}$. This can be due to the JS series coefficients decay much faster to zero for higher order $\chi^n$ terms, alleviating the potential numerical issues of powers of policy ratio, see Table 2. By contrast, the Jeffreys series coefficient decays at the slow rate $n^{-1}$ and could be less preferable.

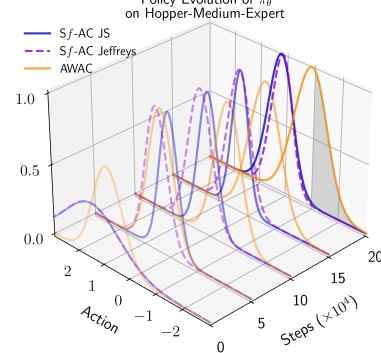

Figure 4: Policy evolution of $Sf$-AC versus AWAC for the first $20\%$ of learning. Minimizing forward KL of AWAC increasingly prompts the policy beyond the minimum allowed action $-1$ (shaded area).

**Clipping Threshold $\epsilon$.** We also compare different $\epsilon$ to see the practical effect of clipping. Figure 5 (B) shows the result on HalfCheetah with $\epsilon = \{0.2, 1, 10, 100\}$. The lower limit $1 - \epsilon$ is set to 0 when negative. The dashed vertical line shows the median performance for $\epsilon = 0.2$ which is a standard value in PPO. In general S$f$-AC is consistent across $\epsilon$.

**Asymmetry vs. Symmetry.** Prior works have shown benefits of symmetric divergence when used in $D_{\mathrm{Opt}}$ it is therefore interesting to ablate the contribution of symmetric $D_{\mathrm{Reg}}$ to clarify when and why this leads to tangible gains. To this end, we compare the following four cases: (1) Asymmetric $D_{\mathrm{Reg}}$, asymmetric $D_{\mathrm{Opt}}$. Specifically we opt for the standard KL divergence for asymmetry. As such, the resulting algorithm can be instantiated by AWAC. (2) Symmetric $D_{\mathrm{Reg}}$, symmetric $D_{\mathrm{Opt}}$. This is the proposed method S$f$-AC, we opt for the Jensen-Shannon. (3) Symmetric $D_{\mathrm{Reg}}$, asymmetric $D_{\mathrm{Opt}}$. This corresponds to using a standard KL loss to approximate the policy induced by Eq. 5. (4) Asymmetric $D_{\mathrm{Reg}}$, symmetric $D_{\mathrm{Opt}}$. In this case, a symmetric loss divergence is adopted to match the KL-regularized policy. This idea underlies the existing works in LLM alignment (Go et al., 2023). As a result, Figure 6 shows the comparison on HalfCheetah medium-expert dataset. It can be seen that symmetric $D_{\mathrm{Reg}}$ perform similarly and outperforms asymmetric $D_{\mathrm{Reg}}$ by a large margin. This suggests that symmetric $D_{\mathrm{Reg}}$ may be preferred over asymmetric ones, regardless of $D_{\mathrm{Opt}}$.

## 7 RELATED WORK

**Taylor expansion in RL.** There are some papers studying the Taylor expansion in RL. Specifically, Tang et al. (2020) proposed to expand the action value difference as an infinite series in the transition dynamics and liken that to the Taylor series. Motoki et al. (2024) proposed to modify the extreme Q-learning objective (Garg et al., 2023) by expanding the exponential function into a MacLaurin series to stabilize learning. In this paper, we utilize the Taylor expansion of $f$-divergence to obtain a $\chi^n$-series and based on it derive analytic policy $\pi^*$ and a tractable minimization objective.

**$f$-divergence in RL.** BRPO requires an analytic $\pi^*$ to be used as the target policy in a loss divergence. The existing literature has mostly focused on asymmetric divergences such as the KL, $\chi^2$, $\alpha$

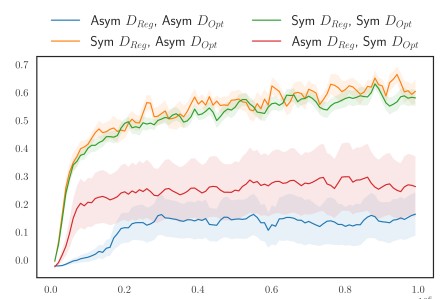

Figure 6: Ablation of symmetric $D_{\mathrm{Reg}}$ on HalfCheetah medexp. Symmetric $D_{\mathrm{Reg}}$ performs similarly and outperforms asymmetric $D_{\mathrm{Reg}}$ by a large margin.

divergences that permit an analytic policy. In other areas such as the goal-conditioned RL (Ma et al., 2022; Agarwal et al., 2023) or RLHF (Go et al., 2023; Wang et al., 2024), symmetric $f$-divergences have been discussed since they require only minimizing the divergence as a loss objective and no $\pi^*$ is required. Their objective can be derived by computing only $f'$ and gradient of log-likelihood.

**Policy ratio clipping.** Clipping policy ratio into a range $[1 - \epsilon, 1 + \epsilon]$ has been widely studied since the proximal policy optimization (Schulman et al., 2017; Vaswani et al., 2022). In the offline context the clipping has been shown to also play a key role (Zhuang et al., 2023). Our method shows that $\epsilon$ can be connected to the convergence requirements of Taylor series. In fact, the clipped $\chi^n$ series shares similarity to the higher order objectives in (Tang et al., 2020) which assumed bounded total deviation, providing an alternate interpretation for their method.

## 8 CONCLUSION

In this paper, we study symmetric divergence regularization for BRPO and show two major issues limiting the use of symmetric divergences: (1) they do not permit an analytic regularized policy, and (2) they can incur numerical issues when naïvely computed. We tackled the two issues by leveraging the finite Taylor series of symmetric divergences, arriving at S$f$-AC, the first BRPO algorithm with symmetric divergences. Through empirical evaluation, we verified that S$f$-AC achieved consistently strong results. Additionally, while the other baselines suffer from weak performances in some environments, S$f$-AC Jensen-Shannon is the only algorithm which was able to consistently rank within top-3 across all the tasks.

## REPRODUCIBILITY STATEMENT

The code is available as a zip file in the supplementary material. We provide detailed experimental settings in Appendix B including the network architectures, hyperparameters and the number of seeds.

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

# Appendix

## A MATHEMATICAL DETAILS AND PROOF

### A.1 PROOF OF THEOREM 1

Let us write out the Lagrangian of the objective (Li et al., 2019; Xu et al., 2023):

$$
\mathcal{L}(\pi, \alpha, \beta) = \sum_s d^{\pi_\mathcal{D}}(s) \sum_a \left[ \pi(a|s)Q(s,a) - \tau\, \mathbb{E}_\mu \left[ f\left( \frac{\pi(a|s)}{\mu(a|s)} \right) \right] \right]
$$
$$
- \sum_s d^{\pi_\mathcal{D}}(s) \left[ \alpha(s) \left( \sum_a \pi(a|s) - 1 \right) - \sum_a \beta(a|s)\,\pi(a|s) \right]. \tag{9}
$$

where $d^{\pi_\mathcal{D}}$ is the stationary state distribution of the behavior policy. $\alpha$ and $\beta$ are Lagrangian multipliers for the equality and inequality constraints, respectively. The KKT conditions are:

Primal feasibility: $\quad \sum_a \pi(a|s) = 1, \quad \pi(a|s) \geq 0,$

Dual feasibility: $\quad \beta(a|s) \geq 0,$

Stationarity: $\quad \dfrac{\partial \mathcal{L}}{\partial \pi(a|s)} = Q(s,a) - \tau\,\mu(a|s) f'\left( \dfrac{\pi(a|s)}{\mu(a|s)} \right) \dfrac{1}{\mu(a|s)} - \alpha(s) + \beta(a|s) = 0,$

Complementarity: $\quad \beta(a|s)\,\pi(a|s) = 0$

Following (Li et al., 2019; Xu et al., 2023), we eliminate $d^{\pi_\mathcal{D}}$ since we assume all policies induce an irreducible Markov chain. For any action with $\pi(a|s) > 0$, we have $\beta(a|s) = 0$. Therefore, we can derive the solution as:

$$
f'\left( \frac{\pi(a|s)}{\mu(a|s)} \right) = \frac{Q(s,a) - \alpha(s)}{\tau} \quad \Rightarrow \quad \pi^*(a|s) = \left[ (f')^{-1}\left( \frac{Q(s,a) - \alpha(s)}{\tau} \right) \right]_+,
$$

where $\alpha(s)$ is the normalization constant that ensures $\pi^*$ sums to 1.

For $\pi^*$ to be analytic, we need to know how to compute $\alpha(s)$. Since $f(t)$ begins with a $t \ln t$ term, so $f'(t) = \ln t + 1 + g'(t)$. If $g'(t)$ is not a function such that $f'(t) = a \ln t + b$ for some constants $a, b$, then $\alpha(s)$ cannot be calculated from the constraint $\sum_a \pi^*(a|s) = 1$.

### A.2 PROOF OF THEOREM 3

To prove Theorem 3, we need to show that (i) the policy expression when $N = 2$ and (ii) when $N > 5$ the solution does not have an analytic expression. To show (i), we study regularization with the sole, general $\chi^n$. Again let us write out the Lagrangian similar to Eq.(9):

$$
\mathcal{L}(\pi, \alpha, \beta) = \sum_s d^{\pi_\mathcal{D}}(s) \sum_a \left[ \pi(a|s)Q(s,a) - \tau \frac{(\pi(a|s) - \mu(a|s))^N}{\mu(a|s)^{N-1}} \right]
$$
$$
- \sum_s d^{\pi_\mathcal{D}}(s) \left[ \alpha(s) \left( \sum_a \pi(a|s) - 1 \right) - \sum_a \beta(a|s)\,\pi(a|s) \right].
$$

where $d^{\pi_D}$, $\alpha$ and $\beta$ are carry the same meaning as Eq.(9). The KKT conditions are now:

Primal feasibility: $\quad \sum_a \pi(a|s) = 1, \quad \pi(a|s) \geq 0,$

Dual feasibility: $\quad \beta(a|s) \geq 0,$

Stationarity: $\quad \dfrac{\partial \mathcal{L}}{\partial \pi(a|s)} = Q(s,a) - \tau \, \dfrac{N \, (\pi(a|s) - \mu(a|s))^{N-1}}{\mu(a|s)^{N-1}} - \alpha(s) + \beta(a|s) = 0,$

Complementarity: $\quad \beta(a|s) \, \pi(a|s) = 0$

Following a similar procedure, we can obtain

$$Q(s,a) - \alpha(s) = \tau \, \frac{N \, (\pi(a|s) - \mu(a|s))^{N-1}}{\mu(a|s)^{N-1}}$$

$$\Rightarrow (\pi(a|s) - \mu(a|s))^{N-1} = \mu(a|s)^{N-1} \frac{(Q(s,a) - \alpha(s))}{N\tau}.$$

$$\Rightarrow \pi^*(a|s) = \mu(a|s) \left[ 1 + \left( \frac{Q(s,a) - \alpha(s)}{N\tau} \right)^{\frac{1}{N-1}} \right]_+ ,$$

where $\alpha(s)$ is the normalization constant ensuring $\sum_a \pi^*(a|s) = 1$. When $N = 2$, this becomes

$$\pi^*(a|s) = \mu(a|s) \left[ 1 + \frac{Q(s,a) - \alpha(s)}{2\tau} \right]_+$$

by redefining $\tau' = 2\tau$ we conclude the proof of (i).

Now let us consider the case where we have $\chi^2$ and $\chi^3$ appearing together, all other KKT conditions remain the same except for the stationarity:

$$\frac{\partial \mathcal{L}}{\partial \pi(a|s)} = Q(s,a) - 2\tau \, \frac{f^{(2)}(1)}{2!} \frac{\pi(a|s) - \mu(a|s)}{\mu(a|s)}$$

$$- 3\tau \, \frac{f^{(3)}(1)}{3!} \left( \frac{(\pi(a|s) - \mu(a|s))}{\mu(a|s)} \right)^2 - \alpha(s) - \beta(a|s) = 0,$$

Now let us define

$$W(a|s) := \frac{\pi(a|s) - \mu(a|s)}{\mu(a|s)}, \quad \tau_2 := 2\tau \, \frac{f^{(2)}(1)}{2!}, \quad \tau_3 := 3\tau \, \frac{f^{(3)}(1)}{3!}$$

$$\Rightarrow \quad W(a|s) = \frac{-\tau_2 + \sqrt{\tau_2^2 + 4\tau_3 \, (Q(s,a) - \alpha(s))}}{2\tau_3}$$

$$\Rightarrow \pi^*(a|s) = \mu(a|s) \left[ 1 + \frac{-\tau_2 + \sqrt{\tau_2^2 + 4\tau_3 \, (Q(s,a) - \alpha)}}{2\tau_3} \right]_+ .$$

Though the reciprocal term becomes more complex, the role it plays still lies in determining the threshold for truncating actions. We can similarly derive the solution for $\sum_{n=2}^{N=4} f^{(n)}(1)\chi^n/n!$ with more complex (and tedious) algebra, but their solutions still take the form $\mu(a|s) \left[ 1 + Z(s,a) \right]_+$, where $Z$ contains the radicals over $Q(s,a)$. As have shown by (Zhu et al., 2023), the truncation effect can be fully controlled by specifying $\tau$. Therefore, we opt for the simplest case where $n = 2$.

The series $\sum_{n=2}^{N} f^{(n)}(1)\chi^n/n!$ is an $N$-th order polynomial in the policy ratio. Therefore, for $N \geq 5$, by the famous Abel-Ruffini theorem (Ramond, 2022) we conclude that it is impossible to have any analytic solution.

### A.3 Proof of Theorem 4

We follow (Barnett et al., 2002, Theorem 1) in proving this result. We start with the following Taylor representation with the integral remainder:

$$f(t) = f(z) + \sum_{n=0}^{N} \frac{(t-z)^n}{n!} f^{(n)}(z) + \frac{1}{N!} \int_z^t (t-z)^N f^{(N+1)}(a) \, da.$$

Specifically by (2.4) of (Barnett et al., 2002) we have

$$
\left| f(t) - f(z) - \sum_{n=0}^{N} \frac{(t-z)^n}{n!} f^{(n)}(z) \right| \le \frac{1}{N!} \left| \int_z^t |t-z|^N \left| f^{(N+1)}(a) \right| \, da \right|
$$

$$
\le \frac{1}{N!} \sup_{a \in [1-\epsilon, 1+\epsilon]} \left| f^{(N+1)}(a) \right| \left| \int_z^t |t-a|^N da \right|
$$

$$
= \frac{1}{(N+1)!} \left\| f^{(N+1)} \right\|_\infty |t-z|^{N+1}
$$

$$
= \frac{1}{(N+1)!} \left\| f^{(N+1)} \right\|_\infty \epsilon^{N+1},
$$

where in the last equation we let $t = \pi^*/\pi_\theta$ clipped to $1 + \epsilon$ and $z = 1$. Now we can repeat the same procedure for $t = 1 - \epsilon$. Since states are sampled from the dataset randomly, we have

$$
\mathbb{E}_{s \sim \mathcal{D}} \left[ \frac{2\epsilon^{N+1}}{(N+1)!} \left\| f^{(N+1)} \right\|_\infty \right] = \sum_s \frac{1}{|\mathcal{D}|} \frac{2\epsilon^{N+1}}{(N+1)!} \left\| f^{(N+1)} \right\|_\infty = \frac{2\epsilon^{N+1}}{(N+1)!} \left\| f^{(N+1)} \right\|_\infty.
$$

We conclude the proof of Theorem 4 by changing $f$ to $g$.

### A.4 Existing Characterizations of $\pi^*$.

We review related work on characterizing the solution of $f$-divergence regularization. They are mainly two ways for characterization, which we discuss in detail below.

| Regularization Characterization | DICE Characterization |
|---|---|
| **(R1)** $\pi^*(a\|s) > 0 \Rightarrow \pi_{\mathcal{D}}(a\|s) > 0$; 
 **(R2)** $h_f(t) := t f(t)$ is strictly convex; 
 **(R3)** $f(t)$ is continuously differentiable. 
 **Result:** 
 $\pi^*(a\|s) \propto \left[ (h_f')^{-1} \left( \frac{Q(s,a)}{\tau} \right) \right]_+.$ | **(D1)** $d^{\pi^*}(s,a) > 0 \Rightarrow d^{\pi_{\mathcal{D}}}(s,a) > 0$; 
 **(D2)** $f(t)$ is strictly convex; 
 **(D3)** $f(t)$ is continuously differentiable. 
 **Result:** 
 $\frac{d^{\pi^*}(s,a)}{d^{\pi_{\mathcal{D}}}(s,a)} \propto \left[ (f')^{-1} \left( \frac{Q(s,a)}{\tau} \right) \right]_+.$ |

#### A.4.1 Regularization Characterization.

We call the first class Regularization Characterization as they exactly studied Eq. (1) (Li et al., 2019; Xu et al., 2023). Here, $\propto$ indicates *proportional to* up to a constant not depending on actions. Assumptions (R2) does not hold for symmetric divergences in general.

**Jeffrey's divergence.** $D_{\text{Jeffrey}}(\pi^* \| \pi_\theta) = D_{\text{KL}}(\pi^* \| \pi_\theta) + D_{\text{KL}}(\pi_\theta \| \pi^*)$ is induced by $f(t) = (t-1)\ln t$. We see that $h_f(t) = (t^2 - t)\ln t$, and therefore $h_f'(t) = (2t-1)\ln t + t - 1$; $h_f''(t) = 2\ln t + 3 - \frac{1}{t}$, which can be negative and in turn indicates that $h_f$ is not strictly convex. Therefore, Jeffrey's divergence does not satisfy their Assumption (R2).

**Jensen-Shannon Divergence.** Recall the Jensen-Shannon divergence is defined by

$$f(t) := t \ln t - (t+1) \ln \frac{t+1}{2}.$$

We examine Assumption (R2) of regularization characterization (Li et al., 2019; Xu et al., 2023):

$$h_f(t) := tf(t) = t^2 \ln t - t^2 \ln \frac{t+1}{2} - t \ln \frac{t+1}{2},$$

$$\Rightarrow h_f'(t) = 2t \ln t + t - 2t \ln \left( \frac{t+1}{2} \right) - \frac{t^2}{t+1} - \ln \left( \frac{t+1}{2} \right) - \frac{t}{t+1},$$

$$\Rightarrow h_f''(t) = 2 \ln \left( \frac{2t}{t+1} \right) + \frac{1}{t+1}.$$

suggesting that $h_f(t)$ is not a convex function and does not meet their Assumption (R2).

**GAN Divergence.** From Table 1 the GAN divergence is defined by

$$f(t) = t \ln t - (t+1) \ln(t+1).$$

Again we focus on its second assumption:

$$h_f(t) = t^2 \ln t - t^2 \ln(t+1) - t \ln(t+1)$$

$$\Rightarrow h_f'(t) = 2t \ln \left( \frac{t}{t+1} \right) - \ln(t+1)$$

$$\Rightarrow h_f''(t) = 2 \ln \left( \frac{t}{t+1} \right) + \frac{1}{t+1}.$$

$h_f''(t)$ can be negative, therefore Assumption (R2) is not satisfied.

### A.4.2 DICE CHARACTERIZATION

DIstribution Correction Estimation (DICE) methods estimate stationary distribution ratios that correct the discrepancy between the data distribution and the optimal policy's stationary distribution (Nachum et al., 2019; Nachum & Dai, 2020).

In the offline context, the optimal solution is the ratio between the stationary distributions $d^{\pi^*}(s,a)/d^{\pi_D}(s,a)$ (Lee et al., 2021; Mao et al., 2024). The optimal policy can then be uniquely identified by $\pi^*(a|s) = d^{\pi^*}(s,a)/\sum_b d^{\pi^*}(s,b)$ (Puterman, 1994). Assumptions (D1)-(D3) can be satisfied by the symmetric divergences in Table 1. However, the issue lies in that $(f')^{-1}$ in general does not have a closed-form expression.

**Jeffrey's divergence.** $f(t) = (t-1) \ln t$, the inverse function of $f'(t) = \ln t + 1 - \frac{1}{t}$ involves the Lambert W function which does not have an analytic expression (Nowozin et al., 2016).

**Jensen-Shannon Divergence.** The generator function of Jensen-Shannon divergence is:

$$f(t) := t \ln t - (t+1) \ln \frac{t+1}{2} \quad \Rightarrow \quad f'(t) = \ln t - \ln \frac{t+1}{2},$$

$$\Rightarrow \pi^*(a|s) \propto \max \left\{ 0, \ (f')^{-1} \left( \frac{Q(s,a)}{\tau} \right) \right\} = \max \left\{ 0, \ \frac{\exp \left( \frac{Q(s,a)}{\tau} \right)}{2 - \exp \left( \frac{Q(s,a)}{\tau} \right)} \right\}.$$

To make sure the policy is a valid distribution, we need to find the normalization constant. However, the integral of $(f')^{-1}$ diverges to infinity, suggesting that no such normalization constant exists.

## B IMPLEMENTATION AND ADDITIONAL RESULTS

### B.1 IMPLEMENTATION DETAILS

We use the MuJoCo suite from D4RL (Apache-2/CC-BY licence) (Fu et al., 2020) for offline experiments. The D4RL offline datasets all contain 1 million samples generated by a partially trained SAC agent. The name reflects the level of the trained agent used to collect the transitions. The Medium dataset contains samples generated by a medium-level (trained halfway) SAC policy. Medium-expert mixes the trajectories from the Medium level and that produced by an expert agent.

| Dataset | S$f$-AC JS | S$f$-AC Jeffreys | AWAC | IQL | XQL | SQL |
|---|---|---|---|---|---|---|
| HalfCheetah-Medium-Expert | 0.001 | 0.001 | 0.0003 | 0.0003 | 0.0002 | 0.0002 |
| HalfCheetah-Medium-Replay | 0.001 | 0.001 | 0.0003 | 0.0003 | 0.0002 | 0.0002 |
| HalfCheetah-Medium | 0.001 | 0.001 | 0.0003 | 0.0003 | 0.0002 | 0.0002 |
| Hopper-Medium-Expert | 0.001 | 0.001 | 0.001 | 0.001 | 0.0002 | 0.0002 |
| Hopper-Medium-Replay | 0.001 | 0.001 | 0.0003 | 0.0003 | 0.0002 | 0.0002 |
| Hopper-Medium | 0.001 | 0.001 | 0.0003 | 0.001 | 0.0002 | 0.0002 |
| Walker2d-Medium-Expert | 0.001 | 0.001 | 0.001 | 0.0003 | 0.0002 | 0.0002 |
| Walker2d-Medium-Replay | 0.001 | 0.001 | 0.0003 | 0.0003 | 0.0002 | 0.0002 |
| Walker2d-Medium | 0.001 | 0.001 | 0.001 | 0.001 | 0.0002 | 0.0002 |

Table 5: The best learning rate across environments. Published settings were used for baselines.

| Dataset | S$f$-AC JS | S$f$-AC Jeffreys | AWAC | IQL | XQL | SQL |
|---|---|---|---|---|---|---|
| HalfCheetah-Medium-Expert | 0.01 | 0.01 | 1.00 | 0.33 | 2.00 | 5.00 |
| HalfCheetah-Medium-Replay | 0.01 | 0.01 | 1.00 | 0.33 | 2.00 | 5.00 |
| HalfCheetah-Medium | 0.01 | 0.01 | 0.50 | 0.33 | 2.00 | 5.00 |
| Hopper-Medium-Expert | 0.01 | 0.01 | 1.00 | 0.33 | 2.00 | 2.00 |
| Hopper-Medium-Replay | 0.01 | 0.01 | 0.50 | 0.33 | 2.00 | 2.00 |
| Hopper-Medium | 0.01 | 0.01 | 0.50 | 0.33 | 2.00 | 5.00 |
| Walker2d-Medium-Expert | 0.01 | 0.01 | 0.10 | 0.33 | 2.00 | 5.00 |
| Walker2d-Medium-Replay | 0.01 | 0.01 | 0.10 | 0.33 | 2.00 | 5.00 |
| Walker2d-Medium | 0.01 | 0.01 | 0.10 | 0.33 | 2.00 | 5.00 |

Table 6: The best $\tau$ across environments. Published settings were used for baselines.

Medium-replay consists of samples in the replay buffer during training until the policy reaches the medium level of performance. In summary, the ranking of levels is Medium-expert > Medium > Medium-replay. The codebase[1] used in this paper is from public repositories (Xiao et al., 2023; Zhu et al., 2025a).

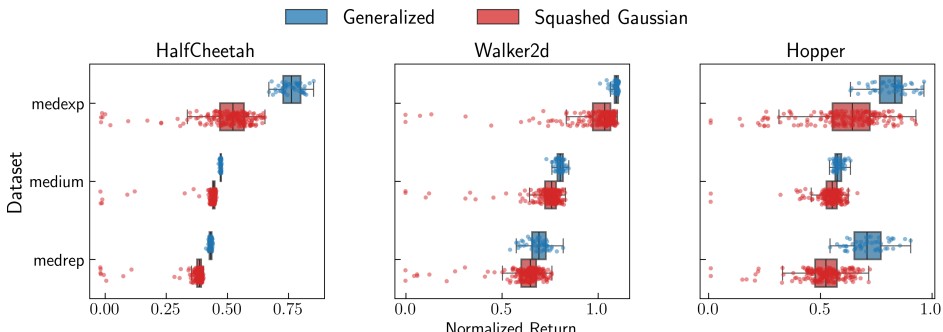

Figure 7: Generalized parametric policy $\pi_\theta$ versus the standard Squashed Gaussian policy. Dots are from the last $50\%$ of learning evaluation. Generalized $\pi_\theta$ can better capture the characteristics of finite-support $\pi^*$ and improves S$f$-AC performance.

**Experiment settings:** We conducted the offline experiment using 9 datasets provided in D4RL: halfcheetah-medium-expert, halfcheetah-medium, halfcheetah-medium-replay, hopper-medium-expert, hopper-medium, hopper-medium-replay, walker2d-medium-expert, walker2d-medium, and walker2d-medium-replay. We run 6 agents: S$f$-AC Jensen-Shannon (JS), S$f$-AC Jeffreys, AWAC, IQL, XQL, and SQL, all with the Squashed Gaussian policy parametrization. Each agent was trained for $1 \times 10^6$ steps. The policy was evaluated every 1000 steps. The score was averaged over 5 rollouts in the real environment; each had 1000 steps.

---

[1]https://github.com/hwang-ua/inac_pytorch
https://github.com/lingweizhu/qexp

| Hyperparameter | Value |
|---|---|
| Learning rate | Swept in $\{3 \times 10^{-3}, 1 \times 10^{-3}, 3 \times 10^{-4}, 1 \times 10^{-4}\}$ 
 See the best setting in Table 5 |
| Temperature | Same as the number reported in 
 the publication of each algorithm. 
 Swept in $\{1.0, 0.5, 0.01\}$. 
 See the setting in Table 6 |
| IQL Expectile | 0.7 |
| Discount rate | 0.99 |
| Hidden size of Value network | 256 |
| Hidden layers of Value network | 2 |
| Hidden size of Policy network | 256 |
| Hidden layers of Policy network | 2 |
| Minibatch size | 256 |
| Adam.$\beta_1$ | 0.9 |
| Adam.$\beta_2$ | 0.99 |
| Number of seeds for sweeping | 5 |
| Number of seeds for the best setting | 10 |

Table 7: Default hyperparameters and sweeping choices.

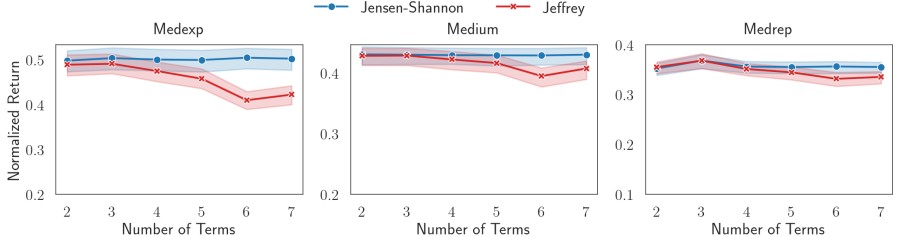

Figure 8: S$f$-AC is insensitive to the number of terms used to compute the symmetry divergence.

**Parameter sweeping:** S$f$-AC results in the paper were generated by the best parameter setting after sweeping. For the baselines, their published settings were used. The best learning rates are reported in Table 5, and the temperatures are listed in Table 6. We list other parameter settings in Table 7.

**Computation Overhead:** All experiments were run on a NVIDIA DGX Station A100 with 128 CPU cores but no GPUs were used. In terms of computation time, S$f$-AC Jensen-Shannon took on average $\approx 10$ hours for 1 million steps, while the Jeffreys took $\approx 6$ hours.

### B.2 Additional Results

By fixing the policy ratio to be $\pi^*(a|s)/\pi_\theta(a|s)$, S$f$-AC avoids the numerical issue when $\pi^*(a|s) = 0$ due to the $q$-exponential. Some papers have reported that utilizing a generalized parametric policy $\pi_\theta$ can significantly improve the performance to capture the characteristics of such finite-support $\pi^*$ (Zhu et al., 2025b). To this end, we run S$f$-AC with the same setting $\epsilon = 100, N_{\text{loss}} = 3$ but with generalized parametric policy.

Figure 7 compares the performance of the generalized parametric policy against the standard Squashed Gaussian. Dots are from the evaluation of the last $50\%$ of learning. It can be seen that a generalized parametric policy indeed significantly improves the performance in terms of median across environment-dataset combinations and greatly reduce variance: low-score red dots do not appear for the generalized policy.

It is visible from Figure 8 that S$f$-AC is insensitive to the number of terms used to compute the symmetry divergence. This is desirable as S$f$-AC avoids the burdensome environment-specific parameter tuning to save resources and gain interpretability.

Figure 9 shows the result on more difficult environments Adroit Pen, Franka Kitchen and Antmaze umaze-diverse. Again the published settings are used for XQL and IQL. It is visible that S$f$-AC

Table 8: Averaged wallclock time (minutes) for S$f$-AC across $N$ and baselines.

| **Jeffrey** $N = 2$ | **JS** $N = 2$ | **JS** $N = 3$ | **JS** $N = 6$ |
|---|---|---|---|
| $364.91 \pm 2.95$ | $365.89 \pm 3.42$ | $329.57 \pm 1.44$ | $333.67 \pm 2.77$ |

| **JS** $N = 7$ | **XQL** | **IQL** | **SQL** |
|---|---|---|---|
| $335.03 \pm 1.61$ | $326.80 \pm 2.01$ | $238.53 \pm 2.18$ | $227.67 \pm 1.72$ |

performs competitively against XQL and IQL and is never the worst across tasks. Figure 10 compares the same algorithms on Adroit Relocate, Hammer, Door "v0" environments, with dataset levels "human" and "cloned". It can be seen that S$f$-AC is on par with or better than the baselines.

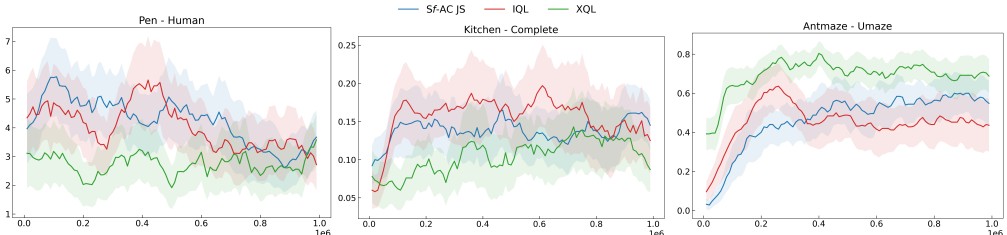

Figure 9: S$f$-AC compares against IQL and XQL on extra environments Adroit Pen, Franka Kitchen and Antmaze umaze-diverse. It is visible that S$f$-AC is never the worst performer across the tasks.

Table 8 lists the wallclock time of S$f$-AC across the number of terms and against the baselines. It is visible that increasing the number of terms does not increase the computation time and S$f$-AC is on the same magnitude as XQL, which takes slightly more time than IQL and SQL. The computation of S$f$-AC does not require storing intermediate results or variables and hence no extra memory is required. The following code snippet computes the series for S$f$-AC Jeffrey, and it is clear that only the resulting sum is needed. All the wallclock time is recorded for 1 million steps using CPU Intel 8457C and GPU Nvidia A6000.

```
1 jeffrey_series = torch.sum(torch.hstack([(-1)**n / n *
2     self.clamp_ratio((ratio - 1)**n) for n in range(2, self.num_terms)]),
3     dim=1, keepdim=True)
```
Listing 1: The conditional symmetry term of Jeffrey divergence.

### B.3 WHY $D_{\text{REG}}$ AND $D_{\text{OPT}}$ SHOULD MATCH

The question why $D_{\text{Reg}}$ and $D_{\text{Opt}}$ should match can be answered by noticing $D_{\text{Reg}}$ defines the regularized problem:

$$
\begin{cases}
\pi^*(a|s) = \arg\max_\pi \mathbb{E}_{\substack{s \sim \mathcal{D} \\ a \sim \pi}} \left[ Q^*(s, a) - D_{\text{Reg}}(\pi(\cdot|s)||\pi_\mathcal{D}(\cdot|s)) \right], \\
\pi_\theta = \arg\min_\theta \mathbb{E}_{s \sim \mathcal{D}} \left[ D_{\text{Opt}}(\pi^*(\cdot|s)||\pi_\theta(\cdot|s)) \right], \\
Q^*(s, a) = r(s, a) + \gamma \mathbb{E}_{\substack{s' \sim \mathcal{D} \\ a' \sim \pi_\theta}} \left[ Q^*(s', a') - D_{\text{Reg}}(\pi_\theta(\cdot|s')||\pi_\mathcal{D}(\cdot|s')) \right],
\end{cases}
\tag{10}
$$

The core argument for setting $D_{\text{Reg}} = D_{\text{Opt}}$ is fixed-point consistency. The overall algorithm seeks a fixed point where $\pi^*$ is consistent with the current $Q$-function, and this consistency is defined by the metric $D_{\text{Reg}}$. If $D_{\text{Opt}} \neq D_{\text{Reg}}$, the resulting $\pi_\theta$ is the optimal policy for a **different optimization problem** than the one defined by $D_{\text{Reg}}$. This introduces an optimization mismatch error, causing the iterative process to converge to a point inconsistent with the true regularized optimal policy $\pi^*$.

**Motivating Example: $D_{\text{Reg}}$ as KL.** Let us assume that $D_{\text{Reg}}$ is the KL divergence. The overall goal is to find a policy $\pi^*$ that is a fixed point of the iterative optimization process. This fixed point is defined by the MaxEnt Bellman Equation (Vieillard et al., 2020), which is intrinsically derived using

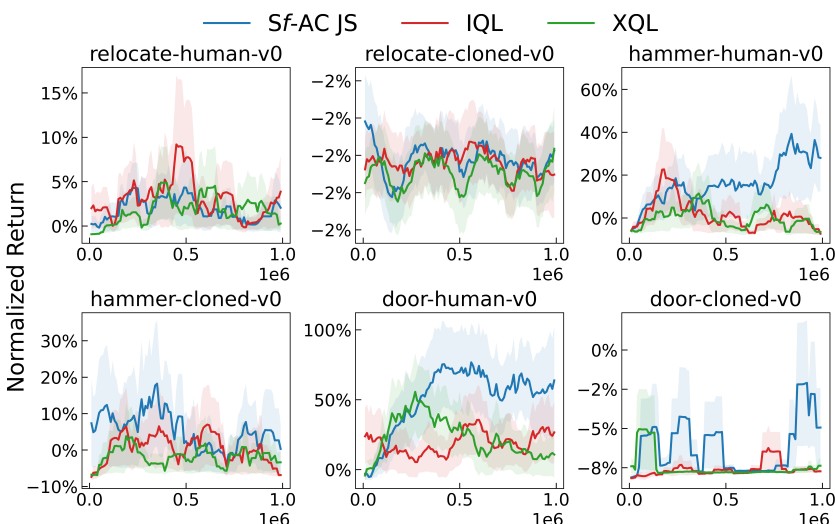

Figure 10: Comparison between S$f$-AC Jensen-Shannon against IQL and XQL on the Adroit relocate, hammer and door "v0" environments. S$f$-AC performs favorably against the baselines.

$D_{\text{Reg}}$. In this case $\pi^*$ is well-defined and given by the Boltzmann-type policy:

$$\pi^*(a|s) \propto \pi_{\mathcal{D}}(a|s) \exp\left(\frac{1}{\tau} Q(s,a)\right),$$

If $D_{\text{Opt}} = D_{\text{Reg}} = D_{\text{KL}}(\pi^*||\pi_\theta)$, the projection step then ensures that the $\arg\min_\theta D_{\text{KL}}(\pi^*||\pi_\theta)$ is a consistent and optimal way to fit the exponential family target $\pi^*$ to the parameterized policy $\pi_\theta$. This ensures the iterative updates are consistent, allowing the algorithm to converge to the true optimal solution $\pi^*$.

**Symmetric $D_{\text{Reg}}$ and $D_{\text{Opt}}$ vs. only symmetric $D_{\text{Opt}}$.** We apply the same analysis to the symmetric regime and compare against the existing works that employ a KL $D_{\text{Reg}}$ but symmetric $D_{\text{Opt}}$ (Go et al., 2023; Wang et al., 2024), this setting is inherited from DPO (Rafailov et al., 2023). If $D_{\text{Opt}} \neq D_{\text{Reg}}$, the resulting policy $\pi_\theta$ is the optimal fit for the target $\pi^*$ according to the metric $D_{\text{Opt}}$, but not $D_{\text{Reg}}$ that defined the original regularization problem. **This means $\pi_\theta$ and the $Q$-function $Q(s,a)$ (last step of Eq. (10)) will be inconsistent with the true regularization objective,** and the fixed point achieved by the mismatched iteration will be biased away from the true solution $\pi^*$, degrading performance and theoretical guarantees. Consider an extreme case where $D_{\text{Opt}} = L_2$, that prioritizes matching the mean but can completely ignore the shape that is defined by $D_{\text{Reg}}$. Mathematically, we can decompose the policy error into two terms:

$$\epsilon_{\text{Total}} = \epsilon_{\text{Rep}} + \epsilon_{\text{Mismatch}}, \tag{11}$$

where $\epsilon_{\text{Rep}}$ is the unavoidable representation error resulting from representing $\pi^*$ that is not in the class $\{\pi_\theta\}$. $\epsilon_{\text{Mismatch}}$ is an avoidable error incurred from the optimality mismatch $D_{\text{Reg}} \neq D_{\text{Opt}}$. In this case, we notice that this optimization mismatch error can be described by the solution $\pi_\theta^{\text{Opt}} = \arg\min_\theta D_{\text{Opt}}(\pi^*||\pi_\theta)$:

$$\epsilon_{\text{Mismatch}} = D_{\text{KL}}\left(\pi^* \,||\, \pi_\theta^{\text{Opt}}\right) - \min_\theta D_{\text{KL}}\left(\pi^* \,||\, \pi_\theta\right), \tag{12}$$

Therefore, the first term describes how much the solution $\pi_\theta^{\text{Opt}}$ deviates from the true optimum given by the KL $D_{\text{Reg}}$, and the second term is the true optimization objective in this setting.

We can generalize the analysis to arbitrary $D_{\text{Reg}} \neq D_{\text{Opt}}$ and characterize $\epsilon_{\text{Mismatch}}$ by:

$$\epsilon_{\text{Mismatch}} = D_{\text{Reg}}\left(\pi^* \,||\, \arg\min D_{\text{Opt}}(\pi^*||\pi_\theta)\right) - \min_\theta D_{\text{Reg}}\left(\pi^* \,||\, \pi_\theta\right), \tag{13}$$

where the first term depicts the far the best fit deviates from the solution to a different divergence $D_{\text{Opt}} \neq D_{\text{Reg}}$. Again this error is avoidable in Eq. (11). It is context-dependent that whether $\epsilon_{\text{Rep}}$ would outweigh $\epsilon_{\text{Mismatch}}$, but we can conclude that $\epsilon_{\text{Total}}$ is strictly smaller if we set $D_{\text{Reg}} = D_{\text{Opt}}$.

