# OpenReview forum: "Symmetric Behavior Policy Optimization"
_ICLR.cc/2026/Conference — Submitted to ICLR 2026_

### Official Review · Reviewer_LZeD · 2025-10-30

**Soundness:** 3
**Presentation:** 3
**Contribution:** 3
**Rating:** 4
**Confidence:** 4

**Summary:**

This paper presents an interesting perspective on behavior regularized policy optimization: using symmetric divergences, instead of asymmetric divergences, as the implementation of the regularization. The difficulty of using a symmetric divergence is twofold: 1) common symmetric divergences do not admit analytical solutions for the optimal policy; and 2) numerical issues can occur when dealing with a finite support distribution. To tackle the challenges, this paper introduces two techniques that are all based on Taylor expansion. First, in order to derive a closed-form optimal policy, they truncate the divergence in the RL objective to the second order; second, they truncate the divergence used in the policy improvement step to improve numerical stability. Some other techniques are also included in the proposed Sf-AC algorithm. The empirical evaluations are conducted on D4RL, and Sf-AC does demonstrate improvement over previous methods like SQL, IQL, and XQL.

**Strengths:**

1. The presentation of the overall idea is clear to me.

2. The literature review is comprehensive. I especially appreciate the discussion in the Appendix, which covers both policy regularization and distribution matching, and elucidates why the problem studied in this paper is unique and has not been addressed by previous literature.

3. The proposed algorithm appears robust and does not require per-task hyperparameter tuning (Tables 4, 5, 6).

**Weaknesses:**

Although the paper briefly discussed the issue with asymmetric divergence in the introduction section, I would say it would be more appropriate to formally define the problems of using asymmetric divergence somewhere between Sections 2 and 3. In lines 51-53, it is unclear to me how symmetric divergence solves the issue of multiple minimum points due to the capacity of the policy function class.

In short, the current version of the draft doesn’t provide sufficient motivation for me to switch from an asymmetric divergence to a symmetric one. I would consider raising my score if the authors can make this clear in the rebuttal.

**Questions:**

1. In equation (8), the first term uses weights proportional to the exponential of the advantage, while in line 310, the optimal policy is defined by using [1 + A/\tau]_+. What causes the mismatch?

2. The value of \epsilon varies from 0.2 to extremely large values like 100. Is it really necessary to include this hyperparameter?

3. Copy-edits: In Algorithm 1, when defining the advantage regression, the subscript of V should be \phi.

---

> ### Author Response · Authors · 2025-11-20
>
> ## Weaknesses:
> > - Although the paper briefly discussed the issue with asymmetric divergence in the introduction section, I would say it would be more appropriate to formally define the problems of using asymmetric divergence somewhere between Sections 2 and 3. In lines 51-53, it is unclear to me how symmetric divergence solves the issue of multiple minimum points due to the capacity of the policy function class.
>
> **Answer**:
> Thank you for the helpful suggestion. We have added a new Section 2.3 to better explain the issue.
> Existing methods [Go et al. 2023; Wang et al. 2024] have opted for symmetric divergences as $D_{Opt}$. We argue this is suboptimal by revisiting the following analysis.
> First, the policy improvement step is performed:
> $$\pi^* =\arg\max_\pi~\mathbb{E}_{a\sim\pi}[Q(s,a)]
>  -\tau \text{Reg}(\pi|\pi_D)$$
>
> Here, $D_\text{Reg}$ directly shapes the optimality condition and the shape of the target distribution itself.
> Therefore, introducing symmetry into $D_\text{Reg}$ changes how the policy improves, changing the weightings of different actions within the support of the behavior dataset.
> However, existing methods still adopt asymmetric $D_{Reg}$ due to the intractability.
> Rather, they have opted for a symmetric loss $D_{Opt}$, which corresponds to the projection step:
> $$\pi_\theta=
> \arg\min_{\pi_\theta} D_{\text{Opt}}(\pi^* || \pi_\theta)$$
> This step does not change the target distribution $\pi^*$, it only determines how the parameteric policy $\pi_\theta$ can approximate it. Therefore, symmetry within $D_\text{Opt}$ can only improve the quality of projection, but it cannot alter the policy improvement dynamics.
> In summary, symmetric $D_{Opt}$ is not consistent with an asymmetric $D_{Reg}$ and does not reflect the intended change on the optimality, policy dynamics, etc. Moreover, this is further compounded by limited function class that can have multiple minima that possess distinct properties.
>
> > - In short, the current version of the draft doesn’t provide sufficient motivation for me to switch from an asymmetric divergence to a symmetric one. I would consider raising my score if the authors can make this clear in the rebuttal.
>
> **Answer:** Our motivation lies in building a complete and consistent framework for using symmetric regularization.
> Prior works [Go et al. 2023; Wang et al. 2024] use symmetric $D_\text{Opt}$ for LLM alignment and have shown improved performance over standard KL.
> However, they have also reported the instability of symmetric $D_{Opt}$.
> We identify a potential cause in this paper:  their optimization objective -- symmetric $D_{Opt}$ -- is only an approximation to the symmetric regularizer $D_{Reg}$ that is intended to change the policy improvement dynamics.
> The inconsistency between symmetric $D_{Opt}$ and asymmetric $D_{Opt}$ is responsible for the instability.
> We address this problem by studying the true symmetric regularizer $D_{Reg} $, providing a theoretical characterization of it, and practical methods for optimizing symmetric regularizers.
> This is needed first to start seeing where they can be beneficial in RL. We have added additional ablations in Fig. 6 of the draft, which highlights the benefits of using a symmetric regularizer for $D_{Reg}$ in place of an asymmetric one.
>
>
> ## Questions:
> > - In equation (8), the first term uses weights proportional to the exponential of the advantage, while in line 310, the optimal policy is defined by using [1 + A/\tau]_+. What causes the mismatch?
>
> **Answer**: this is a typo: we had q-exp in earlier versions of the paper that has $\exp_q A = [1+(q-1)A]^{\frac{1}{q-1}}_{+}$, where $q$ is a rename of $N$. When $q=2$ this recovers line 310. We have removed this redundant notation.
>
> > - The value of \epsilon varies from 0.2 to extremely large values like 100. Is it really necessary to include this hyperparameter?
>
> **Answer**: $\epsilon$ controls the radius of Taylor series convergence. For the main result we showed $\epsilon=0.2$ which is a standard case per PPO. In ablation study we swept across $\epsilon$ up to $100$ only to illustrate the consistency of our method against this clipping.
>
> > - Copy-edits: In Algorithm 1, when defining the advantage regression, the subscript of V should be \phi.
>
> **Answer:** We have fixed the typo.

---

> > ### Comment · Reviewer_LZeD · 2025-11-23
> >
> > I appreciate your response. However, the core rationale behind the adoption of asymmetric regularization D_{reg} remains unclear to me.
> >
> > In the introduction, the author suggests two potential justifications: (1) In line 46, they claim that a symmetric $D_{\text{reg}}$ has not been investigated, which is not a theoretically valid point to justify the benefit of asymmetry. (2) In line 51, the author states that $D_{\text{opt}}$ should match $D_{\text{reg}}$ so that the optimal policy can be well approximated by the parametric policy. However, it is still unclear why this benefit is exclusive to symmetric $D_{\text{reg}}$; in my view, the expressiveness of the policy function class is the more critical factor than the alignment of $D_{\text{reg}}$ and $D_{\text{opt}}$.
> >
> > Aside from these vague claims, the paper lacks a clear theoretical justification for the benefit of asymmetric $D_{\text{reg}}$. The only supporting evidence appears to be the empirical evaluation in Figure 6. Yet, the reported scores raise concerns, as established offline RL methods like IQL, XQL, and AWAC are known to achieve significantly higher performance than what is shown in both Figure 6 and Figure 3 (refer to https://github.com/tinkoff-ai/CORL for a comprehensive evaluation). Given that XQL and AWAC utilize asymmetric $D_{\text{reg}}$ and $D_{\text{opt}}$, the ablation study does not appear to be convincing.
> >
> > Please correct me if I misunderstood anything.

---

> > > ### Author Response · Authors · 2025-11-27
> > >
> > > Dear Reviewer LZeD,
> > >
> > > The discussion deadline is approaching. We appreciate the opportunity to interact with you. We look forward to hearing from you and to address any further concerns you may have.
> > >
> > > Best regards,
> > > Authors

---

> ### Author Response · Authors · 2025-11-25
> **Response to follow-up questions (part 1)**
>
> Thank you for your follow-up questions.
>
> >- In the introduction, the author suggests two potential justifications: (1) In line 46, they claim that a symmetric $D_{Reg}$ has not been investigated, which is not a theoretically valid point to justify the benefit of asymmetry. (2) In line 51, the author states that $D_{Opt} $should match $D_{Reg}$ so that the optimal policy can be well approximated by the parametric policy. However, it is still unclear why this benefit is exclusive to $D_{Reg}$ symmetric; in my view, the expressiveness of the policy function class is the more critical factor than the alignment of $D_{Reg}$ and $D_{Opt}$.
>
> Answer: We have included a new section in Appendix B.3 to clearly state why $D_{Opt}$ should match $D_{Reg}$. The core argument for setting $D_{Reg} = D_{Opt}$ is fixed-point consistency. The overall algorithm seeks a fixed point where $\pi^*$ is consistent with the current $Q$-function, and this consistency is defined by $D_{Reg}$.
>
> Let us now consider the case where $\pi^{\star}$ cannot be perfectly represented by $\pi_{\theta}$.
> If $D_{Reg} \neq D_{Opt}$, e.g., where $D_{Reg}$ is asymmetric and $D_{Opt}$ is symmetric, as the case in [Go et al., 2023],  we can decompose the error of policy into two terms:
> $$\epsilon_{\text{Total}} = \epsilon_{\text{Rep}} + \epsilon_{\text{Mismatch}}$$
> where $\epsilon_{\text{Rep}}$ is the unavoidable representation error, i.e. using a Gaussian to represent a Boltzmann.
> However, $\epsilon_{\text{Mismatch}}$ is the error that arises because of the optimization mismatch that can be avoided:
> $$\epsilon_{\text{Mismatch}} = D_{KL}({\pi^{\star}}||{\arg\min_{\theta} D_{Opt}(\pi^{\star} || \pi_{\theta})}) - \min_{\theta} D_{KL}({\pi^{\star}}||{\pi_{\theta}}).$$
>
> Recall that $D_{Reg}  = D_\text{KL}$, $\epsilon_\text{mismatch}$ measures how far  the best fit to $D_{Opt}$ is away from the optimal solution defined by $D_{Reg}$, that defines the actual regularized target $\pi^*$.
> Notice that this error is avoidable when we choose $D_{Opt} = D_{Reg}$.
>
> >- Aside from these vague claims, the paper lacks a clear theoretical justification for the benefit of a symmetric $D_{Reg}$. The only supporting evidence appears to be the empirical evaluation in Figure 6. Yet, the reported scores raise concerns, as established offline RL methods like IQL, XQL, and AWAC are known to achieve significantly higher performance than what is shown in both Figure 6 and Figure 3 (refer to https://github.com/tinkoff-ai/CORL for a comprehensive evaluation). Given that XQL and AWAC utilize asymmetric $D_{Reg}$ and $D_{Opt}$, the ablation study does not appear to be convincing.
>
> Answer: Our method provides a systematic framework to utilize symmetric divergences that improve on recent works employing only symmetric $D_{Opt}$ [Go et al. 2023, Wang et al. 2024]. In their setting, $D_{Reg}$ is inherently asymmetric (KL) as they inherit DPO. Our analysis exposes that it incurs extra policy error, as the solution to \arg\min D_{Opt} is biased from that of $D_{Reg}$. We propose to remedy it by introducing symmetry to both $D_{Reg}$ and $D_{Opt}$. Appendix B.3 discusses the rationale in detail.
>
> We have accordingly modified lines 52 as:
> > The inconsistency arises because the regularizer $D_{Reg}$  defines a unique regularized optimal policy. If the parametric policy $\pi_{\theta}$ is not optimized with the same divergence, it will be biased towards the optimum of $D_{Opt}$ rathan than  $D_{Reg}$, incurring  extra policy error and altering policy improvement dynamics, see Appendix B.3 for detailed discussion. Therefore, a key motivation of this paper lies in improving existing work [Go et al. 2023; Wang et al. 2024] that have KL as its $D_{Reg}$ but  symmetric divergence as its optimization objective $D_{Opt}$. We achieve this by analyzing how symmetric divergences can be utilized in a principled manner in both $D_{Reg}$ and $D_{Opt}$, bringing maximal consistency.

---

> ### Author Response · Authors · 2025-11-25
> **Response to follow-up questions (part 2)**
>
> **Regarding the reviewer's claim that our baseline performance appears low**:\
> - All our baselines were implemented based on official codebases of several peer-reviewed ICLR-published research papers and training protocols [Xiao et al., ICLR 2023; Zhu et al., ICLR 2025].
> - We have verified that our reproduced scores are consistent with the results reported in those works, following their hyperparameter and evaluation settings without any major modifications. We release our code that can be used to reproduce all our results as the supplementary material.
> - We have added an explicit note in Appendix B.1 explaining this, along with citations and references to the corresponding public codebases.
> - Additionally, we would like to mention that it is expected that different papers have differences in reported performance for the baselines. Many of these differences stem from code-level implementation tricks that are not necessarily part of the base algorithm. Therefore, the results reported in our draft are a valid evaluation of the baselines.
>
>
> **Regarding XQL in ablation study**:\
>  we note that XQL implicitly models the KL regularization in $D_{Reg}$ and only explicitly extracts the policy with advantage regression ($D_{Opt}$). By contrast, all ablation study methods are with explicit regularization and optimization.

---

### Official Review · Reviewer_Gvf5 · 2025-10-31

**Soundness:** 3
**Presentation:** 2
**Contribution:** 2
**Rating:** 4
**Confidence:** 3

**Summary:**

The paper studies symmetric behavior regularization for offline RL within a BRPO-style framework and asks whether symmetric $f$-divergences admit an analytic optimal policy $\pi*$ suitable for target matching.
The authors prove that for common symmetric divergences (Jeffreys, Jensen–Shannon, GAN) no closed-form $\pi^*$ exists and that naively using symmetric losses can be numerically unstable with finite-support policies. They propose a remedy by Taylor expanding $f$-divergences into a $\chi^n$ series, truncating at small order ($N<5$), which yields an analytic surrogate policy (closed form for $N=2$) and a practical algorithm Sf-AC: advantage regression plus a truncated conditional-symmetry term with ratio clipping and a truncation-error bound. Experiments on a Mixture-of-Gaussians fit and 9 D4RL MuJoCo tasks show competitive and failure-robust performance; the JS/Jeffreys variants are frequently top-3 by AUC.

**Strengths:**

Clear theory: explains why standard symmetric divergences do not yield an analytic $\pi^*$ and exposes instability.

Principled workaround: $\chi^n$ expansion recovers a usable surrogate for $N\le 4$; simple closed form for $N=2$.

Practical loss: decomposes into advantage regression + symmetry expansion, with clipped ratios and an error bound.

Empirical robustness: strong results on D4RL with ablations over series order and clipping; resilient under failure cases.

**Weaknesses:**

Approximation vs exact symmetry: truncation introduces bias; the bias/variance/stability trade-off could be analyzed deeper.

Benchmark scope: limited to MuJoCo; harder OOD domains (AntMaze, Kitchen, Adroit-relocate) would stress-test the method.

Behavior-policy access: guidance is light when $\pi_D$ (or density ratios) are poorly estimated.

No end-to-end convergence or error-propagation guarantees under function approximation for the combined objective.

**Questions:**

Sensitivity to series order $N_{\text{loss}}$ and analytic-policy order $N$ (e.g., 2 vs 3–4)? Any task-dependent guidance?

Wall-clock and memory vs $N_{\text{loss}}$ and clipping $\epsilon$?

Robustness when behavior coverage is narrow or multimodal; diagnostics for ratio misestimation?

Could uncertainty signals (critic ensembles) adapt $\epsilon$ or series coefficients to avoid rare failures?

Any caveats for discrete/bounded actions beyond clipping (e.g., projection effects)?

---

> ### Author Response · Authors · 2025-11-20
> **rebuttal**
>
> We thank the reviewer for the careful reading and feedback. Below we address each weakness and question with additional analysis and clarifications that we will incorporate into the final version.
>
> ## Weaknesses:
> > - Approximation vs exact symmetry: truncation introduces bias; the bias/variance/stability trade-off could be analyzed deeper.
>
> **Answer**: You raise a good point. We have now included the missing N=2 condition in Figure 5. $N=2$ corresponds to mere $\chi^2$ divergence regularization, and is overly aggressive in suppressing distribution shift, thereby reducing performance. Meanwhile, $N=3$ is our recommended default. Theorem 4 provides an upper bound on the truncation error, showing that the truncated objective is not far from the exact infinite Taylor series. Empirically, $N=3$ provides the best bias-variance trade-off. From figure 5, we can see that higher $N$ introduces variance from high-order ratio terms without significant bias reduction in Jefferys. However, Jensen-Shannon is mostly agnostic to the changes in $N$, as long as $N\geq3$.
>
>
>
> > - Benchmark scope: limited to MuJoCo; harder OOD domains (AntMaze, Kitchen, Adroit-relocate) would stress-test the method.
>
> **Answer:**
> We have included new experimets on AntMaze, Franka kitchen and Adroit in Figure 9 of Appendix B.2.
> It is visible that the conclusion is consistent with that on Mujoco: S$f$-AC performs stably across tasks of various difficulty and is never the worst performer.
>
> > - Behavior-policy access: guidance is light when $\pi_D$  (or density ratios) are poorly estimated.
>
> **Answer:** Our algorithm does not require access to or estimation of $\pi_D$. In Algorithm 1, $\pi^*$ is estimated using a separate network $\zeta$ trained via advantage regression, and the loss only uses the ratio $\pi_\zeta / \pi_\theta$. This policy ratio needs to stay in the neighborhood of 1 for the series to converge. Therefore, we do PPO-style clipping using a threshold $\epsilon$. This clipping threshold $\epsilon$ provides consistency to ratio estimation errors, and Theorem 4 bounds the error even under misestimation.
>
> > - No end-to-end convergence or error-propagation guarantees under function approximation for the combined objective.
>
> **Answer:** We agree that we do not provide end-to-end convergence guarantees under function approximation for the combined actor-critic objective. Our theoretical analysis focuses on the behavior of our proposed symmetric regularization component itself. Particularly, Theorem 4 shows that the truncation error of our regularizer $D_\text{Reg}$ is within a small uniform bound relative to the exact symmetric divergence, which ensures that the resulting approximation acts as a controlled perturbation of the ideal regularizer. However, this does not imply end-to-end convergence. This issue remains a challenging open-problem and, to the best of our knowledge, similar end-to-end guarantees are not available for any practical offline RL algorithms (IQL, SQL, AWAC, XQL) when used with function approximation.
>
> ## Questions:
> > - Sensitivity to series order and analytic-policy order
> >  (e.g., 2 vs 3–4)? Any task-dependent guidance?
>
> **Answer**: We have updated Figure 5 to include $N=2$, which corresponds to $\chi^2$ regularization. We notice that $N=2$ is too aggressive and reduces performance. Other than this, we notice that $N=3$ generally performs the best across all the tasks that we have tested. So our guidance is to start with $N=3$ for each task.
>
> > - Wall-clock and memory vs and clipping ?
>
> **Answer**: We provide a new Table 9 in Appendix B.2 summarizing the computation time. S$f$-AC is on the same magnitude with baselines like IQL/XQL.
>
> | **Jeffrey** $N=2$ | **JS** $N=2$ | **JS** $N=3$ | **JS** $N=6$ |
> | :---: | :---: | :---: | :---: |
> | 364.91 $\pm$ 2.95 | 365.89 $\pm$ 3.42 | 329.57 $\pm$ 1.44 | 333.67 $\pm$ 2.77 |
> | **JS** $N=7$ | **XQL** | **IQL** | **SQL** |
> | 335.03 $\pm$ 1.61 | 326.80 $\pm$ 2.01 | 238.53 $\pm$ 2.18 | 227.67 $\pm$ 1.72 |
>
> > - Robustness when behavior coverage is narrow or multimodal; diagnostics for ratio misestimation?
>
> **Answer**: Sf-AC has the same limitation as other BRPO offline RL methods when the behavior coverage is very narrow. In such a case, any policy improvement step deviating from the data can become unreliable. No existing method can overcome it without additional assumptions on data coverage. For the multimodal case, the $\pi_\zeta$ network can represent multimodal behavior policy implicitly through advantage weighted regression. The success of projection to $\pi_\theta$ ultimately depends on the type of policy parameterization used to represent it.

---

> ### Author Response · Authors · 2025-11-20
> **rebuttal continued**
>
> > - Could uncertainty signals (critic ensembles) adapt $\epsilon$ or series coefficients to avoid rare failures?
>
> **Answer**: This is an interesting idea. In our implementation, Sf-AC uses fixed ratio clipping $\epsilon$ and fixed number of $\mathcal X$-series terms $N$. In principle, both of these quantities could be adapted based on uncertainty estimates from a critic ensemble. For example, we could reduce $N$ to increase the regularizer strength or tighten the clipping threshold by decreasing $\epsilon$ in states with high variance amongst Q-values. This would act similar to an adaptive trust region. However, designing such a mechanism is likely non-trivial because now we are coupling the critic uncertainty with divergence approximation. If done incorrectly, it could lead to overly conservative updates. However, this is an interesting future direction to extend our proposed algorithm.
>
> > - Any caveats for discrete/bounded actions beyond clipping (e.g., projection effects)?
>
> **Answer**: For discrete action spaces, the symmetric regularizer can be applied without any additional caveats. With the commonly used softmax policy parameterization, $\pi_D$ would have positive probability for every action, which ensures that the policy ratio and the Taylor expansion remain well-behaved.
>
> For bounded action, Sf-AC does not introduce additional projection issues. On the contrary, clipping the ratio $\pi_\zeta / \pi_\theta$ helps prevent large updates, reducing the potential distortion caused by downstream action-clipping.

---

> ### Comment · Reviewer_Gvf5 · 2025-11-21
> **response to authors**
>
> I appreciate the authors’ comprehensive responses. Most of my previous questions and concerns have been addressed in this rebuttal round. The additional comparison experiments and deeper analyses make the results and conclusions more convincing. Accordingly, I increase my overall score from 4 to 6.

---

### Official Review · Reviewer_RJqh · 2025-11-01

**Soundness:** 3
**Presentation:** 3
**Contribution:** 3
**Rating:** 4
**Confidence:** 3

**Summary:**

This paper addresses a clear theoretical and practical gap in behavior-regularized offline reinforcement learning (BRPO).
While existing approaches rely almost exclusively on asymmetric divergences (e.g., KL or *$\chi^2$*) that induce strong mode-seeking bias, this work systematically investigates the use of symmetric *$f$*-divergences as regularizers — an area that has been largely unexplored in prior literature.

The authors prove that symmetric BRPO generally lacks analytic optimal policies due to the non-affine form of *$f'(t)$* in *$\ln t$* and propose a principled approximation based on finite *$\chi$*-series truncation. This yields the Symmetric f-Actor-Critic (Sf-AC) algorithm, which preserves convexity, maintains bounded approximation error, and provides a balanced compromise between mode-seeking and mode-covering behaviors. Empirical results on D4RL MuJoCo benchmarks show consistent robustness and per-environment stability improvements over strong baselines (IQL, SQL, AWAC, XQL).

However, the empirical evaluation lacks sufficient quantitative evidence to clearly demonstrate the effectiveness of the proposed symmetric regularization. While the results suggest improved robustness, the paper does not include concrete numerical comparisons or diagnostic analyses against existing offline RL approaches that would clarify why symmetric regularization helps. In particular, additional metrics illustrating the robustness deficiencies of conventional mode-seeking or mode-covering methods would make the contribution more complete and convincing.

If such quantitative analyses or diagnostic metrics are provided during the rebuttal period, I would be highly willing to raise my score.

**Strengths:**

* **Theoretical novelty**

   * The paper fills a clear theoretical gap in behavior-regularized offline RL by systematically exploring the use of symmetric *$f$*-divergences. This is a novel perspective that extends the well-established BRPO framework beyond its asymmetric KL- or χ²-based formulations and highlights an under-examined dimension of regularization geometry.

* **Practical workaround through principled approximation**

   * The authors introduce a mathematically grounded yet practical approximation via finite χ-series truncation and Taylor expansion of the conditional symmetry term. This approach preserves convexity and boundedness while providing a tractable implementation, making the otherwise intractable symmetric regularization feasible without major instability.

**Weaknesses:**

* **Lack of rigorous comparative analysis**

The paper mainly presents performance plots (e.g., Figure 3) showing only average returns across environments, without quantitative summaries such as mean ± confidence intervals. This makes it difficult to verify whether the improvements are **statistically significant** or whether the baseline implementations are correctly reproduced. In particular, the reported baseline performances on MuJoCo tasks appear unexpectedly low, raising concerns about possible implementation discrepancies or hyperparameter mismatches. Providing detailed numerical tables with variance statistics would make the comparisons more reliable and transparent.

* **Insufficient diagnostic metrics for robustness**

The claim of “robust performance” remains qualitative. In offline RL, robustness is often better captured by additional statistics such as CVaR or worst-case return, rather than mean performance alone. Including such risk-sensitive or tail-distribution metrics would substantially strengthen the claim that symmetric regularization mitigates the brittleness of conventional mode-seeking methods.

* **Need for deeper analysis contrasting mode-seeking and symmetric behaviors**

While prior works have demonstrated the empirical benefits of using symmetric divergences in *$D_{opt}$*, this paper claims that applying symmetry directly to the regularizer *$D_{reg}$* offers additional advantages. Although the paper clearly articulates the motivation for symmetric regularization, it does not sufficiently highlight the limitations of existing mode-seeking regularization that the proposed method intends to address. In offline RL, purely mode-seeking approaches are often regarded as sufficient for ensuring stability and avoiding out-of-distribution (OOD) actions. Introducing more mode-covering behavior through symmetry could, in principle, encourage exploration into unsupported regions of the dataset, potentially harming performance. Therefore, a deeper analysis or empirical evidence clarifying when and why symmetric regularization provides tangible benefits over purely mode-seeking objectives would make the paper more convincing and theoretically grounded.

**Questions:**

* **Quantitative Comparison**

Could the authors provide detailed numerical results (e.g., mean ± confidence intervals) for each baseline, especially on the MuJoCo tasks where baseline performance appears unusually low, to confirm that the reported improvements are statistically significant and reproducible?

* **Robustness Evaluation**

Beyond average returns, can the authors include additional robustness-oriented metrics (e.g., CVaR, percentile or worst-case returns) to substantiate the claim of “robust performance” and demonstrate that symmetric regularization indeed improves stability over mode-seeking methods?

* **Clarification on D_reg vs. D_opt Effectiveness**

Since prior works have already shown benefits of symmetric divergence when used in *$D_{opt}$*, could the authors isolate or ablate the contribution of using symmetry in *$D_{reg}$* to clarify when and why this leads to tangible gains over purely mode-seeking regularization?

---

> ### Author Response · Authors · 2025-11-20
>
> We thank the reviewer for the careful reading and feedback. Below we address each weakness and question with additional analysis and clarifications that we will incorporate into the final version.
>
> ### **Regarding lack of rigorous comparative analysis**
> Our original Figure 3 already accounts for statistical variability by showing the standard error as a shaded region around the mean return. Perhaps this wasn't sufficiently visually distinguishable in the original draft.
>
> To address the reviewer's concerns regarding experimental rigor, we have made the following changes:
>
> 1. Regarding statistical significance:
>     - Table 7 in Appendix B.2 now reports mean ± std across 5 seeds for all baselines and our method on D4RL MuJoCo tasks, complementing the existing plots.
>     - We have added explanation to clarify that the shaded regions represent standard errors in the caption of Figure 3.
>
> 2. Regarding the reviewer's claim that our baseline performance appears low:
>     - All our baselines were implemented based on official codebases of several peer-reviewed ICLR-published research papers and training protocols [Xiao et al., ICLR 2023; Zhu et al., ICLR 2025].
>     - We have verified that our reproduced scores are consistent with the results reported in those works, following their hyperparameter and evaluation settings without any major modifications. We release our code that can be used to reproduce all our results as the supplementary material.
>     - We have added an explicit note in Appendix B.1 explaining this, along with citations and references to the corresponding public codebases.
>     - Additionally, we would like to mention that it is expected that different papers have differences in reported performance for the baselines. Many of these differences stem from code-level implementation tricks that are not necessarily part of the base algorithm. Therefore, the results reported in our draft are a valid evaluation of the baselines.
>
> ### **Regarding insufficient diagnostic metrics for robustness**
> We thank the reviewer for pointing out the ambiguity in our wording. We have changed our term to "consistency" that refers specifically to stability across datasets of varying difficulty and across different environments, to avoid the confusion with "robustness in the risk-sensitive sense", such as CVaR or worst-case returns. We agree that evaluation of risk-sensitivity is an interesting and orthogonal direction, but it is distinct from the consistency we claim to address.
>
> Behavior-regularized offline RL methods often exhibit high task-to-task variance and sensitivity to dataset quality, leading to inconsistent performance. Our experiments demonstrate that symmetric regularization helps stabilize policy improvement across diverse dataset regimes with the same hyperparameters, thereby demonstrating the consistency we intend to demonstrate.
>
> To clarify this, we have revised the draft with the following changes:
> 1. A new Figure 7 is added in the Appendix B.2 showing that Sf-AC performance is stable across different truncation levels $N$, which further supports the claim that symmetric regularization is less sensitive to hyperparameter choices and dataset difficulty.
> 2. Additionally, we have included explanation that our aim is the insensitivity to hyperparameters like $N$ and $\epsilon$ to ease the burdensome environment-wise tuning, rather than risk-sensitive tail behavior.

---

> ### Author Response · Authors · 2025-11-20
> **rebuttal continued**
>
> ### **Regarding the need for deeper analysis contrasting mode-seeking and symmetric behaviors**
>
> We appreciate the reviewer's insightful question regarding the distinction between mode-seeking and symmetric regularization. While the symmetric divergences are often described as mode-covering in the unconstrained density estimation setting, their behavior is fundamentally different when used as the behavior regularization term $D_\text{Reg}$ in BRPO-style offline RL.
>
> In BRPO, the control objective looks like:
> \begin{equation}
>  \max_{\pi} Q(s,a) - \tau {D_\mathrm{Reg}}\~({\pi} ||{\pi_D})
> \end{equation}
>
> Here, the symmetric divergence $D_\mathrm{Reg}$ appears in the form of a penalty, not a matching loss, and it is always evaluated relative to the behavior distribution $\pi_D$. This means that assigning probability mass to unsupported or low-density regions will incur a large divergence penalty for both the asymmetric and symmetric divergences. Therefore, the symmetric divergence regularizer does not encourage the policy to interpolate between modes or to move outside the dataset support. Instead, it locally reshapes the geometry of the updates within the actions already within the dataset's support.
>
> Additionally, our goal in this research is not to show that mode-seeking or mode-covering behavior is better. Rather, our contribution is to make the use of symmetric divergences in $D_\mathrm{Reg}$ tractable for the first time, and to evaluate their effect within the BRPO-style offline RL. The relative benefits of mode-seeking and mode-covering objectives remain an active subject of debate in many recent works [Chan et al., JMLR 2023], which in part motivated our work. Although we do not provide a general theory for when the symmetric regularization is preferable, our experiments demonstrate consistent improvements across all MuJoCo datasets despite their widely differing behavior distributions. The consistent improvement suggests that symmetric regularization indeed benefits the BRPO update, offering consistency benefits without increasing OOD risk. A deeper theoretical analysis of why the mode-covering behavior helps is beyond the scope of this work and is an important direction for future research.
>
> ## **Questions**:
>
> ### - Quantitative comparison
>
> We have added new Table 4 in Section 6.2 to accurately report the mean $\pm$ std. This result confirms the conclusion that symmetric regularization is performing across environments and datasets. See the answer to *Lack of rigorous comparative analysis* above for more details.
>
>
> ### - Robustness evaluation
> See the answer to *Insufficient diagnostic metrics for consistency* above.
>
>
> ### - Clarification on $D_{Reg}$ vs. $D_{Opt}$ Effectiveness
>
> We have included a new ablation study in section 6.3: ablation study to isolate and highlight the contribution of symmetric $D_{Reg}$. It contains the following experiments:
>
> (1) Asymmetric $D_{Reg}$, asymmetric $D_{Opt}$, which can be seen as AWAC.
>
> (2) Symmetric $D_{Reg}$, symmetric $D_{Opt}$. This is the proposed method S$f$-AC.
>
> (3) Symmetric $D_{Reg}$, asymmetric $D_{Opt}$. This method tests the utility of symmetric regularization.
>
> (4) Asymmetric $D_{Reg}$, symmetric $D_{Opt}$. Existing LLM alignment works that use symmetric divergences.
>
> The new result in Figure 6 confirms that the contribution brought by a symmetric $D_{Reg}$ is indeed substantial. This matches the intuition that $D_{Opt}$ is only for projection and less important than $D_{Reg}$, which directly shapes the optimality.
> More specifically, we have the following two steps in BRPO:
>
> *1. Policy improvement step*\
> First, we use the Q-function and the dataset policy $\pi_D$ to find an improved policy using:
>
> $$\pi^{*} = \arg\max_\pi\mathbb{E}_{a\sim\pi}[Q(s,a)]-\tau \mathrm{Reg}(\pi|\pi_D)$$
>
> Here, $D_\mathrm{Reg}$ directly shapes the optimality condition and the shape of the target distribution itself. Therefore, introducing symmetry into $D_\mathrm{Reg}$ changes how the policy improves, changing the weightings of different actions within the support of the behavior dataset. Symmetric divergences at this step were previously intractable, and making this possible is the central contribution of our work.
>
>
> *2. Projection step*\
> Once we have the improved policy, we project it back to a parameteric actor distribution using:
>
> $$\pi_\theta=\arg\min_{\pi_\theta} D_{\text{Opt}}(\pi^*||\pi_\theta)$$
>
> This step does not change the target distribution $\pi^*$, it only determines how the parameteric policy $\pi_\theta$ can approximate it. Therefore, symmetry within $D_\text{Opt}$ can only improve the quality of projection, but it cannot alter the policy improvement dynamics.
>
> **References:**\
> [Xiao et al., ICLR 2023] The In-Sample Softmax for Offline Reinforcement Learning \
> [Zhu et al., ICLR 2025] q-exponential Family for Policy Optimization\
> [Chan et al., JMLR 2023] Greedification operators for policy optimization: Investigating forward and reverse kl divergences

---

> ### Author Response · Authors · 2025-11-27
>
> Dear Reviewer RJqh,
>
> The discussion deadline is approaching, we appreciate the opportunity to interact with you. We look forward to hearing from you and to address any further concerns you may have.
>
> Best regards,
> Authors

---

### Author Response · Authors · 2025-11-21
**global response to all reviewers**

We thank all reviewers for their careful reading and helpful feedback. We have modified our paper according to the provided comments. Below is a list of the changes we made:

- (Reviewer RJqh) modified caption of Figure 3 to explicitly account for mean and confidence intervals
- (Reviewer RJqh) new Figure 6 that does the ablation study to highlight the contribution of symmetric $D_{Reg}$, and the new paragraph Asymmetry vs. Symmetry in Section 6.2.
- (Reviewer RJqh) new Figure 8 in Appendix B.2 that shows the consistency of the proposed method across $N$.
- (Reviewer Gvf5) new Figure 5 that includes $N=2$
- (Reviewer Gvf5) new Table 8 in Appendix B.2 that compares the wallclock time of different methods.
- (Reviewer Gvf5) new Figure 9 that compares the proposed method on AntMaze, Franka Kitchen and Pen.
- (Reviewer Gvf5) new Figure 10 that compares the proposed method on Adroit Relocate, Hammer and Door.
 - (Reviewer LZeD) a new subsection 2.3 to highlight the drawback of existing approachs
- (Reviewer LZeD) a new appendix section Appendix B.3 that explains the need for setting $D_\text{Reg} = D_\text{Opt}$
- (Reviewer LZeD) explicitly stating our motivation in improving the inconsistency of existing works in lines 52-58
- (Reviewer LZeD) typos and undefined notations fixed

Please refer to the updated paper draft for detail. Again we thank all reviewers for their efforts to make the paper better. We look forward to discussion and to address any remaining concerns you might have.

---

### Comment · Area_Chair_TA3j · 2025-11-25

Dear reviewers:

The authors have submitted their rebuttal, and we now require your follow-up assessments to move the decision process forward. Please review the authors’ responses and update your evaluations accordingly.

Your prompt follow-up is necessary for us to finalize the meta-review.

Kindly submit your updates as soon as possible.

Best,

Area Chair

---

### Meta-Review · Area_Chair_qaRH · 2026-01-06

**Summary:**

This submission studies symmetric behavior regularization for offline RL in a BRPO-style framework. The paper contributes:
(1) theory showing that common symmetric $f$-divergences (e.g., Jeffreys / Jensen--Shannon / GAN-type) typically do not yield analytic optimal policies in the BRPO policy-improvement step; and
(2) a practical approximation via truncated series / Taylor expansions, leading to the proposed Sf-AC algorithm with ratio clipping and an approximation-error bound.
Despite these strengths, my suggested decision is Reject. The main factors are:
(a) persistent uncertainty about evaluation reliability and comparability (including concerns that baseline performances appear unusually low relative to widely reported results and unified benchmark suites); and
(b) an insufficiently compelling and broadly convincing justification that symmetric regularization in $D_{\mathrm{reg}}$ provides a principled advantage over standard asymmetric choices beyond the presented ablations.
While the rebuttal adds tables/ablations and clarifies several points, the remaining concerns materially reduce confidence in the empirical and conceptual conclusions.

**Reviewer Concerns:**

Quantitative reporting and added experiments: The rebuttal adds mean $\pm$ std tables, clarifies that shaded regions correspond to variability estimates, expands ablations (including symmetry placement in $D_{\mathrm{reg}}$ vs. $D_{\mathrm{opt}}$), and broadens benchmarks beyond MuJoCo (e.g., AntMaze / Kitchen / Adroit in the appendix). These changes partially address requests for stronger quantitative evidence and broader coverage.
Sensitivity and practical guidance: The rebuttal provides additional guidance and analyses on truncation order and clipping/sensitivity, and reports compute overhead, addressing concerns about approximation trade-offs and practicality.
Clarifications and corrections: The rebuttal fixes noted typos and clarifies mismatched expressions, and adds discussion intended to motivate divergence-consistency across steps.
Concerns still outstanding
Baseline comparability / protocol alignment: A key concern remains that reported baseline scores (and thus the magnitude of gains) appear misaligned with widely cited results and more comprehensive evaluation suites. The rebuttal’s explanation that different papers use different implementation details is plausible, but it does not fully establish that comparisons are conducted under a community-standard, clearly aligned evaluation protocol. This leaves open the possibility that improvements are substantially protocol-dependent.
Core motivation and justification for symmetric $D_{\mathrm{reg}}$: Although the rebuttal adds a fixed-point consistency argument, at least one reviewer remains unconvinced that matching $D_{\mathrm{reg}}$ and $D_{\mathrm{opt}}$ is the decisive principle compared to function-class expressiveness and other factors. The central rationale for why symmetric $D_{\mathrm{reg}}$ is necessary/beneficial still does not appear universally compelling.
Robustness claims remain narrow: The rebuttal re-scopes “robustness” to “consistency,” which clarifies the claim, but does not provide the risk-sensitive or tail-focused diagnostics originally requested (e.g., worst-case / CVaR). This contributes to an overall sense that the empirical story is not fully settled.
No end-to-end guarantees under function approximation: The paper does not provide end-to-end convergence/error-propagation guarantees for the combined objective under function approximation. While common in offline RL, this remains a limitation.

**Reviewer Scores:**

Reviewer RJqh: likely $4$ to $5$ due to added mean $\pm$ std reporting and additional ablations, but with lingering doubts about baseline alignment and robustness diagnostics.
Reviewer Gvf5: $4$ to $6$ (explicitly stated after the rebuttal).
Reviewer LZeD: likely remains $4$ (no change), as the follow-up indicates unresolved concerns about the core justification and baseline/evaluation mismatch.
Overall, the post-rebuttal assessment remains mixed: one clear supporter, one slightly improved borderline, and one reviewer remaining at/below the acceptance threshold with a fundamental concern.

---

### Decision · Program_Chairs · 2026-01-26

Reject